# Statistics Caching Test-Time Adaptation for Vision-Language Models

**Zenghao Guan**[1,2,3], **Yucan Zhou**[4], **Wu Liu**[5], **Xiaoyan Gu**[1,2,3*]

[1]Institute of Information Engineering, Chinese Academy of Sciences,
[2]School of Cyber Security, University of Chinese Academy of Sciences,
[3]State Key Laboratory of Cyberspace Security Defense,
[4]Tianjin University, [5]University of Science and Technology of China
`zenghaoguan.cs@gmail.com, guxiaoyan@iie.ac.cn`

## Abstract

Test-time adaptation (TTA) for Vision-Language Models (VLMs) aims to enhance performance on unseen test data. However, existing methods struggle to achieve robust and continuous knowledge accumulation during test time. To address this, we propose Statistics Caching test-time Adaptation (SCA), a novel cache-based approach. Unlike traditional feature-caching methods prone to forgetting, SCA continuously accumulates task-specific knowledge from all encountered test samples. By formulating the reuse of past features as a least squares problem, SCA avoids storing raw features and instead maintains compact, incrementally updated feature statistics. This design enables efficient online adaptation without the limitations of fixed-size caches, ensuring that the accumulated knowledge grows persistently over time. Furthermore, we introduce adaptive strategies that leverage the VLM's prediction uncertainty to reduce the impact of noisy pseudo-labels and dynamically balance multiple prediction sources, leading to more robust and reliable performance. Extensive experiments demonstrate that SCA achieves compelling performance while maintaining competitive computational efficiency. The code is available at this link.

## 1 Introduction

Recently, large-scale vision-language models (VLMs), such as CLIP [1], have shown strong generalization across a wide range of tasks, thanks to pre-training on billions of image-text pairs from the web. These models learn powerful cross-modal representations, enabling impressive zero-shot performance without task-specific supervision [1, 2, 3, 4]. However, despite their success, adapting VLMs to real-world tasks remains challenging. Most existing approaches rely on a small number of labeled samples from the target domain for fine-tuning [5, 6, 7], which limits their use in scenarios where annotation is expensive or unavailable [8, 9, 10, 11]. This motivates the need for test-time adaptation (TTA) for VLMs, where the model adapts to new domains on the fly using only unlabeled test data, improving robustness without extra labeling cost [12, 13, 14, 15, 16].

Existing TTA methods for vision-language models (VLMs) generally fall into two categories: (1) test-time prompt tuning methods and (2) cache-based methods. Test-time prompt tuning [12, 13, 17, 18, 19] adapts the prompt to each individual sample in the test data stream using an entropy minimization objective on randomly augmented samples. To avoid prompt degradation caused by errors in previous unsupervised optimization steps, these methods typically process each test sample independently and reset the model after evaluating it. These approaches, while simple, fail to leverage

---

*Corresponding author is Xiaoyan Gu.

39th Conference on Neural Information Processing Systems (NeurIPS 2025).

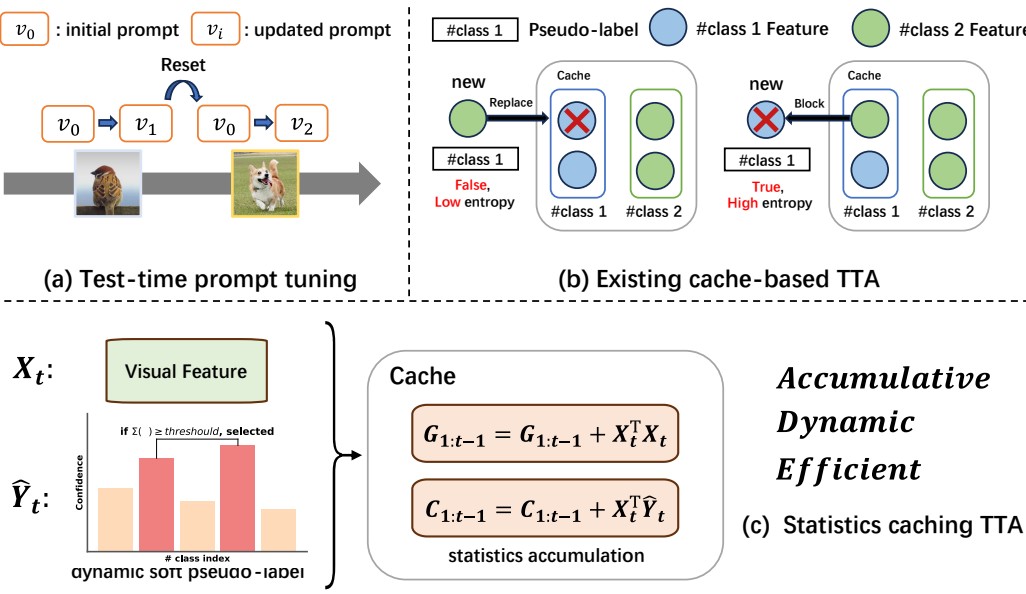

Figure 1: **Comparison of our SCA (c) with test-time prompt tuning methods (a) and existing cache-based methods (b).** We use each test sample's feature and soft pseudo-label to update the cached feature statistics, enabling continuous knowledge accumulation without forgetting.

information from previously seen samples. As a result, the model may repeatedly forget useful patterns, which reduces its ability to adapt and generalize across the test set. Moreover, test-time tuning methods require backpropagation through the text encoder at every test-time training step to learn the text prompt, leading to substantial computational overhead.

Another type of work is cache-based methods [14, 20, 15, 21]. These methods use a class-wise feature cache to retain information during inference. Since ground-truth labels are unavailable, test samples are assigned to cache slots based on zero-shot pseudo-labels. To improve cache quality under limited capacity, low-entropy samples are retained while high-entropy ones are discarded.

Cache-based methods have garnered significant attention due to their potential for knowledge accumulation and computational efficiency. However, a critical question arises: is this acclaimed knowledge accumulation process truly robust and effective over time? Since low entropy does not guarantee correctness, confidently incorrect samples may enter and persist in the cache. Previous studies have primarily highlighted this detrimental impact on knowledge quality, often reflected in decreased classification accuracy. However, they often overlook a more insidious consequence: whether and how such errors fundamentally disrupt and hinder the process of accumulating new knowledge. Here, we analyze the issue from two perspectives:

Firstly, a misclassified low-entropy sample can overwrite a previously correct entry in the cache, leading to the **forgetting** of useful knowledge that had already been acquired. To demonstrate this effect, we conducted a controlled experiment on the Flowers102 [22] using a SOTA cache-based method, DPE [14]. Specifically, we construct a controlled setup with two cache update orders: (1) using only a set of standard samples, and (2) using the same set followed by a few deliberately selected low-entropy misclassified samples that are guaranteed to enter the cache. The second sequence simulates the injection of "forgetting triggers," allowing us to measure the performance drop by directly comparing the performance before and after forgetting. As shown in Fig. 2, DPE suffers from forgetting, which causes a decline in performance as useful information is overwritten.

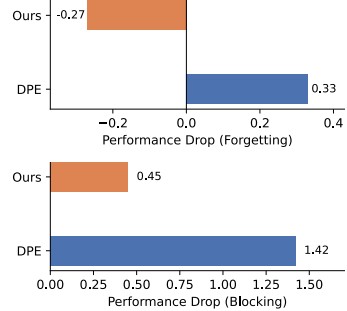

Figure 2: Performance drop on Flowers using ViT-B/16. Negative values mean the performance improves instead of drops.

Secondly, a misclassified low-entropy sample, once admitted, might persistently occupy a cache slot due to its high confidence score. This can **block** potentially correct but higher-entropy samples from entering the cache, thereby

hindering the intended knowledge accumulation and refinement. To demonstrate this, we simulated a 'bad case' scenario, where several low-entropy but incorrect samples were placed at the beginning of the online test sample sequence. As depicted in Fig.2, under these conditions, the cache was consistently dominated by incorrect features, preventing effective knowledge accumulation and resulting in a significant performance drop for DPE compared to a standard evaluation order.

The preceding analysis reveals that while existing TTA methods attempt to adapt, they face significant hurdles in achieving robust and continuous knowledge accumulation. Test-time prompt tuning resets for each sample, inherently discarding prior information and preventing any meaningful accumulation of knowledge from the test stream. Existing cache-based methods, while aiming for accumulation, are susceptible to forgetting due to limited cache capacity and can suffer from blocked accumulation when noisy feature-label pairs lead to the entrenchment of incorrect information. This highlights a critical need for an approach that can consistently learn from all past samples and is robust to pseudo-label noise.

To this end, we propose Statistics Caching test-time Adaptation (SCA), a simple yet effective cache-based test-time approach for VLM. As illustrated in Fig. 1, SCA is designed to effectively accumulate task-specific knowledge from all previously encountered test samples without the forgetting issues inherent in feature-caching methods. We achieve this by modeling the effective use of previous test sample features as a least squares problem. Crucially, solving this problem requires only the corresponding feature statistics, not the full set of historical raw features. Therefore, unlike prior cache-based methods that store a limited subset of raw features, SCA maintains and incrementally updates these compact feature statistics in the cache. This core design choice enables the continuous integration of information from every incoming sample, addressing the problem of forgetting due to fixed-size feature caches and ensuring that the accumulated knowledge grows persistently over time.

Furthermore, to ensure the quality of this continuously accumulated knowledge, especially in the presence of noisy pseudo-labels, we introduce a dynamic soft pseudo-label assignment strategy. This strategy intelligently utilizes the uncertainty information from the VLM's initial predictions to generate more robust pseudo-labels for updating the feature statistics. Finally, we propose an instance-level adaptive fusion strategy that combines the cache (statistics-derived) logit and the textual (zero-shot) logit using prediction entropy. This strategy estimates the confidence of the statistics-derived classifier and adaptively adjusts its weight relative to the VLM's textual logits. As a result, it enables more robust and flexible predictions with relatively little hyperparameter tuning effort.

We evaluate SCA on 15 diverse datasets across out-of-domain and cross-domain benchmarks, where it achieves notable performance gains over state-of-the-art prompt tuning and cache-based methods. Moreover, SCA avoids time-consuming backpropagation and achieves over $10\times$ computational efficiency compared to TPT [12], making it much more practical for resource-limited settings. Our contributions are summarized as follows:

- We propose Statistics Caching test-time Adaptation (SCA), a novel cache-based method that stores feature statistics instead of raw features, enabling effective accumulation of task-specific knowledge from all previously seen test samples without forgetting.

- We introduce a dynamic soft pseudo-labeling strategy that leverages uncertainty to reduce error accumulation. Additionally, we propose an instance-level adaptive fusion mechanism based on prediction entropy, enabling robust predictions without extensive hyperparameter tuning.

- Extensive experimental results show that SCA achieves strong performance across 15 diverse datasets, matching or surpassing state-of-the-art methods while maintaining high computational efficiency.

## 2  Preliminaries

**Zero-shot classification in CLIP.** CLIP [1] enables zero-shot image classification via pre-trained visual $\Phi_I(\cdot)$ and textual $\Phi_T(\cdot)$ encoders mapping inputs to a shared $D$-dimensional space $\mathbb{R}^D$. To classify an image $I$ into one of $K$ classes $\{c_1, ..., c_K\}$, text prompts $T_k$ are generated for each class (e.g., `"a photo of [CLS]"`). The image and prompts are embedded:

$$f_I = \Phi_I(I) \quad \text{and} \quad f_{T_k} = \Phi_T(T_k) \quad \text{for } k = 1, ..., K. \tag{1}$$

Here, $f_I$ and $f_{T_k}$ are the image and text features, respectively. CLIP computes the cosine similarity $\text{cosine}(f_I, f_{T_k})$ between the image feature and each text feature. These similarities determine class

probabilities using a softmax function with temperature $\kappa$:

$$p(c_k|I) = \frac{\exp\left(\text{cosine}(f_{T_k}, f_I)/\kappa\right)}{\sum_{j=1}^{K} \exp\left(\text{cosine}(f_{T_j}, f_I)/\kappa\right)}. \tag{2}$$

The image $I$ is assigned the class $c_k$ corresponding to the maximum probability $p(c_k|I)$.

**Test-time prompt tuning.** These methods [12, 13, 23, 17, 18, 19] optimize learnable prompts for each test sample. As ground-truth labels are unavailable during testing, the optimization is guided by the prediction entropy of selected sample augmentations:

$$\min_{v} \mathcal{L}_(v; I) = \min_{v} - \sum_{k=1}^{K} p(\hat{y} = c_k \mid \mathcal{A}(I), v) \log p(\hat{y} = c_k \mid \mathcal{A}(I), v), \tag{3}$$

where $v$ denotes the learnable prompt, $\mathcal{A}(I)$ represents the set of selected augmentations with lower prediction entropy for test sample $I$, and $p(\hat{y} = c_k \mid \mathcal{A}(I), v)$ denotes the averaged predicted probability over these augmentations.

**Cache-based test-time adaptation.** These methods [14, 20, 15, 21] maintain a fixed-size cache of $M$ entries per class to store features from test samples. This cache acts as an external memory, allowing the model to reference past examples during inference. By continuously updating the cache based on minimal-entropy filtering strategy, the model can gradually accumulate useful information and improve prediction consistency. Final predictions are obtained by combining zero-shot logits from the textual classifier with similarity scores from the cache:

$$\mathbf{z}_k = \alpha_0 \mathbf{z}_{\text{text}} + \alpha_1 \mathbf{z}_{\text{cache}}$$

$$= \alpha_1 f_I^\top f_{T_k} + \alpha_2 \sum_{i=1}^{M} \exp(-\beta(1 - f_I^\top f_i^{(k)})), \tag{4}$$

where $\mathbf{z}_{\text{cahce}}$ denotes the logit for class $k$, and $f_i^{(k)}$ is the $i^{th}$ feature of category $k$ in the cache. Their performance relies heavily on fusion hyperparameters $\alpha_1$ and $\alpha_2$.

## 3 Statistics Caching Test-time Adaptation

To avoid forgetting of previously learned knowledge, we aim to make full use of past test samples observed up to time step $t$ to guide model adaptation. Denote $\boldsymbol{X}_{1:t} \in \mathbb{R}^{n_{1:t} \times d}$ as the features extracted from all test samples up to time $t$, and $\boldsymbol{Y}_{1:t} \in \mathbb{R}^{n_{1:t} \times K}$ as their corresponding one-hot encoded ground-truth labels. Here, $n_{1:t}$ represents the cumulative number of test samples up to time $t$. We make full use of historical samples by formulating it as a least squares problem:

$$\min_{\boldsymbol{W}_t} \|\boldsymbol{X}_{1:t}\boldsymbol{W}_t - \boldsymbol{Y}_{1:t}\|_{\text{F}}^2 + \gamma\|\boldsymbol{W}_t\|_{\text{F}}^2. \tag{5}$$

Here, we use $\gamma$ to control the strength of the regularization. The closed-form solution of Eq. 5 is:

$$\boldsymbol{W}_t = (\boldsymbol{X}_{1:t}^\top \boldsymbol{X}_{1:t} + \gamma\boldsymbol{I})^{-1}\boldsymbol{X}_{1:t}^\top \boldsymbol{Y}_{1:t}. \tag{6}$$

$\boldsymbol{W}_t$ denotes the classifier learned by leveraging the feature representations of all previously observed test samples up to time $t$. It captures accumulated knowledge across test instances, enabling more informed and robust predictions.

However, in practical test-time scenarios, storing all historical sample features $\boldsymbol{X}_{1:t}$ is typically infeasible due to increasing storage overhead or potential privacy concerns. Moreover, ground-truth labels $\boldsymbol{Y}_{1:t}$ are not accessible during testing. Existing methods typically rely on CLIP's zero-shot predictions to assign pseudo-labels for cache construction. However, these pseudo-labels are often noisy, leading to the inclusion of unreliable feature-label pairs that degrade the quality of the updated classifier. To tackle these challenges, we propose the following parts.

**Statistics Accumulation.** In fact, computing $\boldsymbol{W}_t$ does not require storing the growing feature matrix $\boldsymbol{X}_{1:t}$ as test samples accumulate. Instead, it is sufficient to maintain two compact feature statistics that are independent of the sample size: $\boldsymbol{G}_{1:t} = \boldsymbol{X}_{1:t}^\top \boldsymbol{X}_{1:t} \in \mathbb{R}^{d \times d}$ and $\boldsymbol{C}_{1:t} = \boldsymbol{X}_{1:t}^\top \boldsymbol{Y}_{1:t} \in \mathbb{R}^{d \times K}$. Gram matrix $\boldsymbol{G}_{1:t}$ represents the second-order feature statistics, capturing the pairwise relationships between the features, and $\boldsymbol{C}_{1:t}$ represents the first-order statistics, which is essentially the weighted

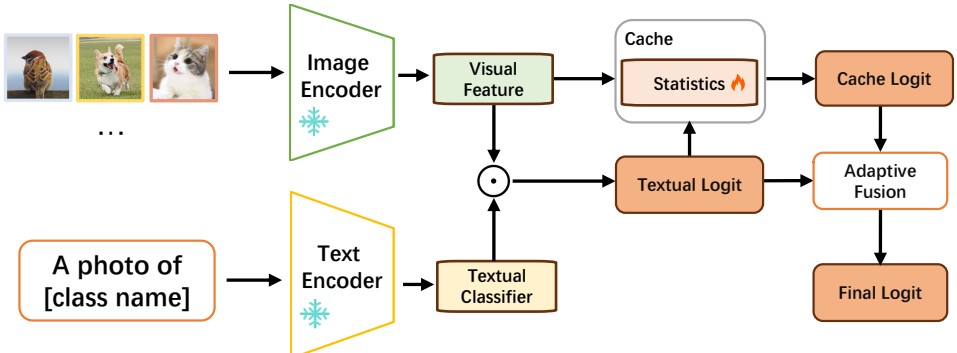

Figure 3: Framework of our SCA. Each test sample updates the feature statistics in the cache. Based on these statistics, we build a closed-form classifier that captures knowledge from all test samples. Finally, we apply a instance-level adaptive fusion strategy to combine the textual and cache logits.

sum of the features and the labels. Due to the linear characteristics of these feature statistics, we can update them additively to support online updates during test-time scenarios:

$$\boldsymbol{G}_{1:t} = \boldsymbol{G}_{1:t-1} + \boldsymbol{X}_t^\top \boldsymbol{X}_t, \quad \boldsymbol{C}_{1:t} = \boldsymbol{C}_{1:t-1} + \boldsymbol{X}_t^\top \boldsymbol{Y}_t. \tag{7}$$

Here, $\boldsymbol{X}_t \in \mathbb{R}^{n_t \times d}$ and $\boldsymbol{Y}_t \in \mathbb{R}^{n_t \times K}$ denote the features of the current test sample and its corresponding label, respectively. Finally, these two statistics are then used to compute $\boldsymbol{W}_t$ as follows:

$$\boldsymbol{W}_t = (\boldsymbol{G}_{1:t} + \gamma \boldsymbol{I})^{-1} \boldsymbol{C}_{1:t}. \tag{8}$$

This approach efficiently accumulates knowledge and builds a strong classifier that captures information from all previously seen samples.

**Dynamic soft pseudo-label assignment.** The above discussion assumes access to true labels when accumulating feature statistics. In practice, we typically rely on pseudo-labels generated by CLIP's zero-shot predictions to update $\boldsymbol{C}_{1:t}$. A common but problematic practice [14, 15] is to convert these predictions into hard pseudo-labels by selecting the single most confident class as ground truth. This approach ignores the nuance of the full probability distribution, collapsing all uncertainty into a binary decision, and can amplify errors when the top prediction is only marginally more probable or even incorrect [24]. A continuous stream of such errors will inevitably skew the learned cache classifier and degrade its accuracy.

To mitigate this issue, we dynamically generate the soft pseudo-label $\hat{\boldsymbol{Y}}_t$ for each test sample $\boldsymbol{X}_t$ based on its predicted class probabilities $\mathbf{p}_{\text{text}}$ from the CLIP textual (zero-shot) classifier. Specifically, we first identify a candidate set of classes based on each sample. Without loss of generality, we assume the class probabilities $\mathbf{p}_{\text{text}} = (\mathbf{p}_{\text{text}}^{(1)}, \mathbf{p}_{\text{text}}^{(2)}, ..., \mathbf{p}_{\text{text}}^{(K)})$ are sorted in descending order. In this case, $\mathbf{p}_{\text{text}}^{(1)}$ denotes the probability of the class initially deemed most likely by the zero-shot classifier. We select the smallest number of classes, denoted by $\Omega$, such that their cumulative probability meets or exceeds a threshold $\tau$:

$$\Omega = \min \left\{ k \ \middle| \ \sum_{k=2}^{K} \mathbf{p}_{\text{text}}^{(k)} \geq \tau (1 - \mathbf{p}_{\text{text}}^{(1)}) \right\} \cup \{1\}. \tag{9}$$

Then the soft pseudo-label $\hat{\boldsymbol{Y}}_t$ is generated as:

$$\hat{\boldsymbol{Y}}_t^{(k)} = \frac{\exp \left( \mathbf{p}_{\text{text}}^{(k)} \cdot \mathbb{I}(k \in \Omega) \right)}{\sum_{l \in \Omega} \exp \left( \mathbf{p}_{\text{text}}^{(l)} \right)}. \tag{10}$$

By allowing $|\Omega|$ vary with $\tau$ and the sample's confidence, our method produces softer pseudo-labels for uncertain samples and sharper labels when the model is confident. This dynamic assignment strategy better captures the sample-specific uncertainty inherent in the zero-shot predictions and effectively curbs the overconfidence often associated with hard labeling schemes. Finally, the pseudo-label $\hat{\boldsymbol{Y}}_t$ is used to update the first-order statistics $\hat{\boldsymbol{C}}_{1:t}$ as shown in Eq.7. The cache classifier $\hat{\boldsymbol{W}}_t$ is then computed as Eq.8 to obtain the cache logits $\mathbf{z}_{\text{cache}} = \boldsymbol{X}_t \hat{\boldsymbol{W}}_t$.

Table 1: **Experimental results on the cross-domain benchmark with two backbones of CLIP.** The best and second-best results are shown **in bold** and underlined, respectively.

| Method | Aircraft | Caltech | Cars | DTD | EuroSAT | Flower | Food101 | Pets | SUN397 | UCF101 | Average |
|---|---|---|---|---|---|---|---|---|---|---|---|
| CLIP-ResNet-50 | 15.66 | 85.88 | 55.70 | 40.37 | 23.69 | 61.75 | 73.97 | 83.57 | 58.80 | 58.84 | 55.82 |
| Ensemble | 16.11 | 87.26 | 55.89 | 40.37 | 25.79 | 62.77 | 74.82 | 82.97 | 60.85 | 59.48 | 56.63 |
| CoOp [5] [IJCV'22] | 15.12 | 86.53 | 55.32 | 37.29 | 26.20 | 61.55 | 75.59 | 87.00 | 58.15 | 59.05 | 56.18 |
| CoCoOp [26] [CVPR'22] | 14.61 | 87.38 | 56.22 | 38.53 | 28.73 | 65.57 | 76.20 | **88.39** | 59.61 | 57.10 | 57.23 |
| TPT [12] [NeurIPS'22] | 17.58 | 87.02 | 58.46 | 40.84 | 28.33 | 62.69 | 74.88 | 84.49 | 61.46 | 60.82 | 57.66 |
| DiffTPT [13] [ICCV'23] | 17.60 | 86.89 | **60.71** | 40.72 | 41.04 | 63.53 | **79.21** | 83.40 | 62.72 | 62.67 | 59.85 |
| TDA [14] [CVPR'24] | 17.61 | 89.70 | 57.78 | 43.74 | 42.11 | 68.74 | 77.75 | 86.18 | 62.53 | 64.18 | 61.03 |
| DPE [15] [NeurIPS'24] | 19.80 | **90.83** | 59.26 | 50.18 | 41.67 | 67.60 | 77.83 | 85.97 | 64.23 | 61.98 | 61.93 |
| BCA [27] [CVPR'25] | **19.89** | 89.70 | 58.13 | 48.58 | **42.12** | 66.30 | 77.19 | 85.58 | 63.38 | 63.51 | 61.44 |
| SCA (Ours) | 20.82 | 91.03 | 59.73 | 53.31 | 43.33 | 71.09 | 76.51 | 86.15 | 64.50 | 66.40 | 63.29 |
| CLIP-ViT-B/16 | 23.67 | 93.35 | 65.48 | 44.27 | 42.01 | 67.44 | 83.65 | 88.25 | 62.59 | 65.13 | 63.58 |
| Ensemble | 23.22 | 93.55 | 66.11 | 45.04 | 50.42 | 66.99 | 82.86 | 86.92 | 65.63 | 65.16 | 64.59 |
| CoOp [5] [IJCV'22] | 18.47 | 93.70 | 64.51 | 41.92 | 46.39 | 68.71 | 85.30 | 89.14 | 64.15 | 66.55 | 63.88 |
| CoCoOp [26] [CVPR'22] | 22.29 | 93.79 | 64.90 | 45.45 | 39.23 | 70.85 | 83.97 | 90.46 | 66.89 | 68.44 | 64.63 |
| TPT [12] [NeurIPS'22] | 24.78 | 94.16 | 66.87 | 47.75 | 42.44 | 68.98 | 84.67 | 87.79 | 65.50 | 68.04 | 65.10 |
| DiffTPT [13] [ICCV'23] | 25.60 | 92.49 | 67.01 | 47.00 | 43.13 | 70.10 | **87.23** | 88.22 | 65.74 | 62.67 | 65.47 |
| TDA [14] [CVPR'24] | 23.91 | 94.24 | 67.28 | 47.40 | 58.00 | 71.42 | 86.14 | 88.63 | 67.62 | 70.66 | 67.53 |
| DPE [15] [NeurIPS'24] | **28.95** | 94.81 | 67.31 | 54.20 | 55.79 | 75.07 | 86.17 | 91.14 | 70.07 | 70.44 | 69.40 |
| Dynaprompt [23] [ICLR'25] | 24.33 | 94.32 | 67.65 | 47.96 | 42.28 | 69.95 | 85.42 | 88.28 | 66.32 | 68.72 | 65.52 |
| BCA [27] [CVPR'25] | 28.59 | 94.69 | 66.86 | 53.49 | 56.63 | 73.12 | 85.97 | 90.43 | 68.41 | 67.59 | 68.59 |
| SCA (Ours) | 28.50 | **94.85** | **68.49** | **57.09** | **57.16** | **76.09** | 86.09 | **91.44** | **70.27** | **73.43** | **70.34** |

**Instance-level Adaptive Fusion.** Previous methods [25, 15, 20] fuse zero-shot (textual) logit and cache logit using a fixed dataset-level coefficient, which typically requires extensive hyperparameter tuning on the validation set or even the test set. In contrast, we propose an instance-level fusion strategy that leverages entropy to estimate classifier confidence and adaptively weights the logits, enabling more robust and adaptive predictions.

For each test sample, we compute the prediction entropy for each classifier and denote them as $\mathcal{H}_{\text{cache}}$ and $\mathcal{H}_{\text{text}}$. Lower entropy indicates higher confidence, so we derive confidence weights as follows:

$$[\alpha_1, \alpha_2] = \text{softmax}\big(-\beta\left[\mathcal{H}_{\text{cache}}, \mathcal{H}_{\text{text}}\right]\big), \tag{11}$$

where $\beta$ controlling the sharpness of the distribution. This enables the model to evaluate their relative confidence, relying more on the zero-shot model when the cache model is uncertain and shifting toward the cache model as its confidence increases. Entropy is used as it is the most common confidence metric. We then compute the aggregated logits as:

$$\mathbf{z}_{\text{final}} = \alpha_1 \mathbf{z}_{\text{cache}} + \alpha_2 \mathbf{z}_{\text{text}} \tag{12}$$

By automatically and adaptively adjusting the fusion coefficient for each test sample, our method captures instance-level uncertainty and improves robustness across diverse data distributions. We also provide an option to amplify the cache logits using the same sharpness parameter $\beta$ in a divisive manner. Using default settings ($\beta = 0.5$) gives strong performance across multiple datasets. In other words, a single set of hyperparameters can be applied to fuse predictions across different datasets, reducing the effort required to tune the fusion parameters $\alpha_1, \alpha_2$.

## 4 Experiments

### 4.1 Experimental Settings

**Datasets**. Following prior work [12, 13, 15, 27], we evaluate our method using two established benchmarks: the cross-domain benchmark and the out-of-distribution (OOD) benchmark. (1) The cross-domain benchmark assesses the model's ability to generalize across various domains, each with its own set of classes. This benchmark includes 10 diverse image classification datasets from distinct domains: FGVCAircraft [28], Caltech101 [29], StanfordCars [30], DTD [31], EuroSAT [32],

Table 2: **Experimental results on the OOD benchmark with two backbones of CLIP.** The best and second-best results are shown **in bold** and underlined, respectively.

| Method | ImageNet | ImageNet-A | ImageNet-V2 | ImageNet-R | ImageNet-S | Average | OOD Average |
|---|---|---|---|---|---|---|---|
| CLIP-ResNet-50 [1] | 58.16 | 21.83 | 51.41 | 56.15 | 33.37 | 44.18 | 40.69 |
| Ensemble | 59.81 | 23.24 | 52.91 | 60.72 | 35.48 | 46.43 | 43.09 |
| CoOp [5] [IJCV'22] | 63.33 | 23.06 | 55.40 | 56.60 | 34.67 | 46.61 | 42.43 |
| CoCoOp [26] [CVPR'22] | 62.81 | 23.32 | 55.72 | 57.74 | 34.48 | 46.81 | 42.82 |
| TPT [12] [NeurIPS'22] | 60.74 | 26.67 | 54.70 | 59.11 | 35.09 | 47.26 | 43.89 |
| DiffTPT [13] [ICCV'23] | 60.80 | **31.06** | 55.80 | 58.80 | 37.10 | 48.71 | 45.69 |
| TDA [14] [CVPR'24] | 61.35 | 30.29 | 55.54 | 62.58 | 38.12 | 49.58 | 46.63 |
| DMN-ZS [20] [CVPR'24] | **63.87** | 28.57 | 56.12 | 61.44 | 39.84 | 49.97 | 46.49 |
| DPE [15] [NeurIPS'24] | 63.41 | 30.15 | **56.72** | **63.72** | **40.03** | 50.81 | **47.66** |
| BCA [27] [CVPR'25] | 61.81 | 30.35 | 56.58 | 62.89 | 38.08 | 49.94 | 46.98 |
| SCA (Ours) | 63.11 | 30.64 | 56.51 | 62.85 | 39.33 | 50.49 | 47.33 |
| CLIP-ViT-B/16 [1] | 66.73 | 47.87 | 60.86 | 73.98 | 46.09 | 59.11 | 57.20 |
| Ensemble | 68.34 | 49.89 | 61.88 | 77.65 | 48.24 | 61.20 | 59.42 |
| CoOp [5] [IJCV'22] | 71.51 | 49.71 | 64.20 | 75.21 | 47.99 | 61.72 | 59.28 |
| CoCoOp [26] [CVPR'22] | 71.02 | 50.63 | 64.07 | 76.18 | 48.75 | 62.13 | 59.91 |
| TPT [12] [NeurIPS'22] | 68.98 | 54.77 | 63.45 | 77.06 | 47.94 | 62.44 | 60.81 |
| DiffTPT [13] [ICCV'23] | 70.30 | 55.68 | 65.10 | 75.00 | 46.80 | 62.28 | 60.52 |
| TDA [14] [CVPR'24] | 69.51 | 60.11 | 64.67 | 80.24 | 50.54 | 65.01 | 63.89 |
| DMN-ZS [20] [CVPR'24] | **72.25** | 58.28 | 65.17 | 78.55 | **53.20** | 65.49 | 63.80 |
| DPE [15] [NeurIPS'24] | 71.91 | 59.63 | **65.44** | 80.40 | 52.26 | 65.93 | 64.43 |
| Dynaprompt [23] [ICLR'25] | 69.61 | 56.17 | 64.67 | 78.17 | 48.22 | 63.37 | 61.81 |
| BCA [27] [CVPR'25] | 70.22 | **61.14** | 64.90 | 80.72 | 50.87 | 65.37 | 64.16 |
| SCA (Ours) | 71.75 | 60.33 | 65.38 | **80.85** | 52.50 | **66.16** | **64.77** |

Flowers102 [22], Food101 [33], OxfordPets [34], SUN397 [35], and UCF101 [36]. (2) The OOD benchmark measures the model's robustness against natural shifts in data distribution. It evaluates performance on ImageNet [37] and four challenging variants: ImageNet-A [9], ImageNet-V2 [10], ImageNet-R [38], and ImageNet-Sketch [39].

**Implementation Details**. Consistent with prior work [15, 25], our experiments adopt ResNet-50 [40] and ViT-B/16 [41] as the visual encoders for CLIP, with a batch size of 1 to satisfy online processing requirements. For textual prompts, we employ the prompt ensembling strategy as used in previous works [15, 42, 20, 25]. For data augmentation, we adopt the approach from DPE [12], generating 63 randomly resized crops per test image for the OOD benchmark. No data augmentation is applied in the cross-domain benchmark. By default, we set the ridge coefficient $\gamma$ to 1e4, threshold $\tau$ to 0.1, sharpness coefficient $\beta$ to 0.5. All experiments are conducted on a single NVIDIA A100 RTX GPU, using top-1 accuracy to measure classification performance.

**Baselines**. We compare our method with zero-shot CLIP using either a simple prompt or prompt ensemble, few-shot methods such as CoOp [5], with 16-shot annotated samples per class, alongside the following established CLIP test-time adaptation approaches: (1) prompt-tuning methods: TPT [12], DiffTPT [13], Dynaprompt[23]; (2) cache-based methods: TDA [14], DMN-ZS [20], DPE [15]. We also include recent baseline BCA [27] that adapts likelihood using Bayes rules. To ensure a fair comparison, we exclude RLCF [43] and COSMIC [21], which rely on auxiliary strong models (e.g., ViT-L-14) during test-time adaptation. We also exclude some transductive TTA baselines [44, 45] that rely on utilizing inter-sample relations within a large batch. The results of these baselines are directly taken from their respective original papers.

### 4.2 Results and Discussions

**Cross-Domain Benchmark**. Table 1 presents results on the cross-domain benchmark. While CoOp [5] and CoCoOp [26] improve CLIP's generalization, they rely on training-based adaptation with labeled data, which restricts their practical applicability. In contrast, our SCA outperforms CoOp by an average of 7.11% and 5.71%, CoCoOp by an average of 6.06% and 6.46% on two backbones in a training-free manner, demonstrating superior generalization without relying on labeled examples. Compared to test-time prompt tuning methods, SCA achieves a 4.82% gain in average accuracy over the strongest baseline, Dynaprompt [23]. It also consistently outperforms cache-based approaches, surpassing TDA [14] by 2.25% and 2.81%, and DPE [15] by 1.36% and 0.94% on ResNet-50 and ViT-B/16 backbones, respectively. Notably, unlike existing cache-based methods [15, 14] that rely

on dataset-specific fusion hyperparameters $(\alpha_1, \alpha_2)$ for optimal performance, our method uses the same configuration for all datasets within a benchmark-backbone pair, while different benchmarks or backbones use different configurations, highlighting its strong generalization.

**OOD Benchmark**. In Table 2, we further evaluate the generalization capability of our proposed method on the OOD benchmark by comparing it with state-of-the-art approaches. Our proposed method consistently outperforms the CoOp [5] and CoCoOp [26] with substantial improvements across all datasets. Specifically, it improves average accuracy by about 5% on both backbones. These results demonstrate the strong generalization capability of our approach and its effectiveness in enhancing performance without any additional supervised training. On the ResNet-50 backbone, our method performs slightly below the strongest baseline DPE [15]; however, it achieves this level of performance without any training overhead, unlike DPE, which requires additional training. In contrast, on the ViT-B/16 backbone, SCA consistently outperforms all existing methods, with average accuracy improvements ranging from 0.23% to 3.72% across various OOD datasets.

**Computation Efficiency.** We measure time per image for several baselines on SUN397 (19,850 samples) using a single 80 GB NVIDIA RTX A100 GPU as an example of computational efficiency. Table 3 presents the corresponding results. Since our proposed SCA avoids time-consuming backpropagation, it offers a notable advantage in computational efficiency compared to prompt tuning methods. Specifically, our approach is over 10× faster than TPT [12]. Compared to the cache-based methods TDA [14], our approach is slightly less efficient but delivers notable performance gains, especially on the cross-domain benchmark.

Table 3: **Computation efficiency comparison on SUN397 using ViT-B/16.**

| Method | BP-free | Time Per Image (s) |
|---|---|---|
| CLIP [1] | ✓ | 0.007 |
| TPT [12] | ✗ | 0.218 |
| TDA [14] | ✓ | 0.009 |
| DPE [15] | ✗ | 0.043 |
| **SCA (Ours)** | ✓ | 0.012 |

**Storage Overhead.** Previous cache-based methods that store feature-label pairs with fixed size $M$ have a storage cost of $M \times K \times d$. Our cached feature statistics accumulate knowledge from all samples and require $(K + d) \times d$ storage. While this may be less efficient when the number of classes $K$ is small, our method offers lower storage overhead as $K$ grows, for example on large-scale datasets like ImageNet.

## 4.3 Ablation Studies

**Effectiveness on statistics accumulation and dynamic soft label assignment.** To investigate the roles of our statistics accumulation and dynamic soft label assignment, we report the performance on 10 fine-grained recognition datasets involved in cross-domain datasets using ViT-B/16 in Table 4. [W/o statistics accumulation] means that we follow [14, 15] by using a fixed-size cache of $M$ feature-label pairs per class to build our classifier (Eq. 6). [W/o dynamic soft label assignment] means that we use hard pseudo-labels for statistics accumulation. We

Table 4: **Effectiveness on statistics accumulation and dynamic soft label assignment.** We report the averaged results on 10 datasets in cross-domain benchmark.

| Method | Avg |
|---|---|
| w/o statistics accumulation ($M = 4$) | 69.31 |
| w/o statistics accumulation ($M = 8$) | 69.47 |
| w/o statistics accumulation ($M = 16$) | 69.75 |
| w/o statistics accumulation ($M = 32$) | 69.72 |
| w/o dynamic soft label assignment | 69.32 |
| with both | **70.34** |

have the following key observations: (1) Replacing statistics accumulation with direct feature caching leads to a performance drop compared to our method. While increasing the cache size gradually improves performance, the gains eventually plateau. Interestingly, when the cache size reaches 8, it outperforms the state-of-the-art baseline DPE [15] (69.40). This improvement stems from our use of second-order feature information (the Gram matrix $G$) to construct the cache-based classifier, whereas prior cache-based methods rely solely on first-order information. Studies in other areas of machine learning similarly conclude that second-order feature statistics significantly enhance performance [46, 47, 48, 49]. (2) Using hard labels to update statistics leads to a performance drop. This highlights the importance of the extra information in soft labels and demonstrates the effectiveness of our dynamic label assignment.

**Ablations on fusion coefficients.** In our SCA, the fusion coefficient is automatically adjusted for each test sample. We perform an ablation study by manually setting a fixed fusion coefficient, varying $\alpha_1$ from 0.1 to 0.9. As shown in Fig. 4, the optimal fusion value differs across datasets. While the

adaptive coefficients may not be optimal for every case, they consistently outperform most fixed coefficients across different datasets. Compared to some cache-based methods [14, 15] that require costly parameter searches to tune fusion hyperparameters for each dataset, our approach uses a single set of hyperparameters for all datasets to fuse the prediction and still deliver strong performance. This achieves a good trade-off between performance and tuning effort.

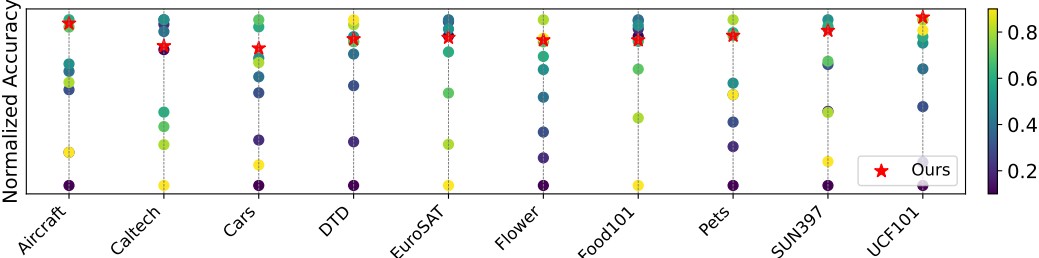

Figure 4: Normalized accuracy with varying fusion parameters on ViT-B/16.

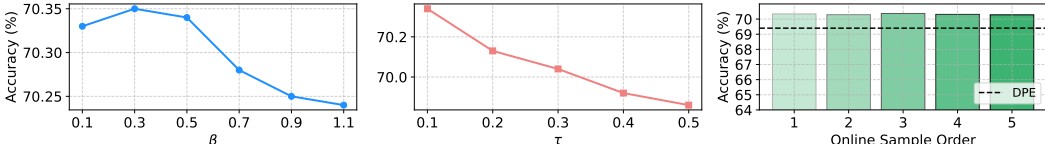

Figure 5: **Left:** hyperparameter analysis of sharpness coefficient $\beta$. **Middle:** hyperparameter analysis of threshold $\tau$. **Right:** sensitivity analysis of test-time sample order.

**Hyperparameter Analysis.** Here, we present the results of hyperparameter analysis for the threshold $\tau$ and the sharpness coefficient $\beta$ on 10 fine-grained recognition datasets. As shown in Fig. 5 (Left and Middle), the performance remains relatively stable when the parameters are set within a reasonable range. This suggests that our method is not sensitive to these hyperparameters, making it more robust and easier to apply in practice.

**Sensitivity to test time sample order.** Since our SCA is designed for online test-time adaptation, its performance may be affected by the order in which test samples are encountered. To assess this sensitivity, we conduct experiments on 10 datasets from the cross-domain benchmark, each with varying test-time sample orders. As shown in Fig. 5 (Right), our method demonstrates stable performance across different orders and consistently outperforms the state-of-the-art baseline DPE [15], highlighting its robustness to input sequence variations in online settings.

## 5 Related Work

**Test-time prompt tuning**. These methods tune the text prompts using the test samples. TPT [12] optimizes the prompt by minimizing prediction entropy to promote consistency across augmented views of the same input. DiffTPT [13] follows the TPT's framework and uses a pre-trained diffusion model to generate more diverse augmentations. C-TPT [18] optimizes prompts by minimizing calibration error, while RLCF [43] introduces an additional CLIP model as a reward signal to guide test-time prompt tuning. Despite their effectiveness, they typically treat each test sample in isolation, resetting the model for every instance without leveraging information from previously seen samples. To effectively accumulate previously learned knowledge, it is essential to incorporate a cache module. For instance, Dynaprompt [23] uses a prompt buffer and introduces a dynamic selection strategy to update the appropriate prompt stored in the buffer. Although this approach facilitates the accumulation of historical knowledge, it, like other test-time tuning methods, requires backpropagation through the text encoder at each test-time step, resulting in significant computational overhead.

**Cache-based test-time adaptation.** These methods store high-confidence visual features during test time and utilize them to provide extra information for current prediction. TDA [14] and DMN [20] establish visual caches from test data and pseudo-labels to retrieve class prototypes during test time. Similarly, DPE [15] maintains a dynamic cache that facilitates the training of residual dual prototypes. COSMIC [21] enhances the semantic diversity of pseudo-labels by leveraging fine-grained visual features from DINOv2. These methods demonstrate high computational efficiency and

strong performance due to the rich information provided by features stored in the cache. However, they suffer from forgetting and blocking issues during knowledge accumulation. Additionally, fusing the predictions derived from the cache features with the VLM's zero-shot predictions often requires extensive and careful tuning. Our SCA overcomes these challenges through statistics accumulation combined with dynamic label assignment and fusion strategy.

## 6 Conclusion

We propose Statistics Caching test-time Adaptation (SCA), a cache-based method that stores feature statistics instead of raw features, allowing effective accumulation of task-specific knowledge without forgetting. We also introduce a dynamic soft pseudo-labeling strategy that uses uncertainty to reduce error accumulation, and an instance-level adaptive fusion mechanism based on prediction entropy for robust predictions without heavy hyperparameter tuning. Experiments on 15 diverse datasets show that SCA matches or exceeds state-of-the-art performance with high computational efficiency.

While our method enables test-time accumulation of task-specific knowledge without training and achieves strong performance, it still shares some limitations with existing methods. Firstly, storing feature statistics introduces some storage overhead. Secondly, our method only mitigates error accumulation during test time to a certain extent, and the performance still falls short of the ideal setting where ground-truth labels are available to build the classifier. Moreover, similar to existing CLIP TTA methods, our approach mainly focuses on the single-domain scenario, where all test samples come from a single dataset. Extending CLIP-based TTA to the multi-domain setting where test samples from multiple domains are mixed is an important yet underexplored direction. In such cases, naive accumulation may even lead to negative effects due to domain discrepancies. Since feature representations from different domains often exhibit distinctive patterns, a promising direction is to identify each sample's domain and accumulate feature statistics separately for each domain. In future work, we plan to further investigate more effective ways to leverage feature statistics and develop improved strategies for robust knowledge accumulation across domains.

## Acknowledgements

This work is supported by the Chinese Academy of Sciences under grant No. XDB0690302, the National Key Research and Development Program of China (NO. 2024YFE0203200), and the National Nature Science Foundation of China (NO. U24A20329).

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

# Statistics Caching Test-Time Adaptation for Vision-Language Models

## Appendix

## A   Details on forgetting and blocking issues

Since low-entropy zero-shot predictions are not always correct (as shown in the following figure), they can lead to forgetting and blocking issues that hinder the accumulation of new knowledge in existing cache-based methods [14, 15, 20]. We conduct experiments to illustrate this (Figure 2 in the Introduction). Below, we present the experimental settings and analysis in detail.

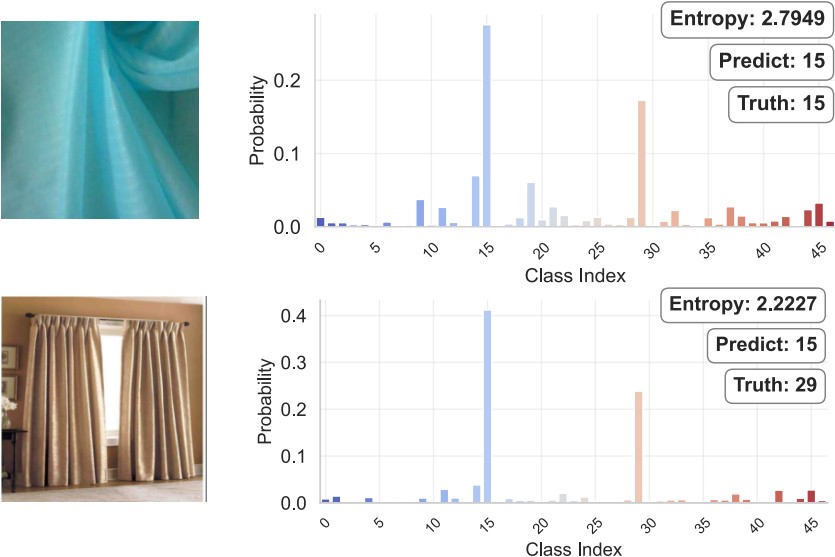

Figure A1: An example of wrong high confidence sample.

**Forgetting:** a misclassified low-entropy sample can overwrite a previously correct entry in the cache, leading to the forgetting of useful knowledge that had already been acquired. To better illustrate forgetting issues, we conduct an additional set of control experiments. Firstly, we identify a small set of "forgetting trigger samples." For each dataset, we randomly select 10 classes and choose the single lowest-entropy misclassified sample for each class. To ensure these 10 trigger samples are admitted by the cache, we filter out any correctly classified samples from the entire test set that have even lower entropy. From the remaining pool of test samples, we randomly draw 500 samples to serve as our fixed "final test samples." All other remaining samples are designated as "initial samples," which allow the cache to accumulate sufficient class-specific knowledge. We then consider the following two sequences for updating the model's cache:

- Sequence A: `[initial samples]`
- Sequence B: `[initial samples, forgetting trigger samples]`

The forgetting trigger samples, with their low entropy, are designed to infiltrate the cache built from the initial samples, causing it to forget previously learned information. In other words, the cache updated with Sequence A represents the state before forgetting, while the cache updated with Sequence B represents the state after forgetting. To quantify the impact of forgetting, we measure the performance difference between Sequence A and Sequence B on the final test samples, where the performance drop is defined as the accuracy of Sequence A minus the accuracy of Sequence B. The results are summarized in the following figure.

We have several observations: (1) The SOTA baseline DPE is highly vulnerable to forgetting. (2) SCA is inherently robust to forgetting, as its accumulation design prevents error samples from fully

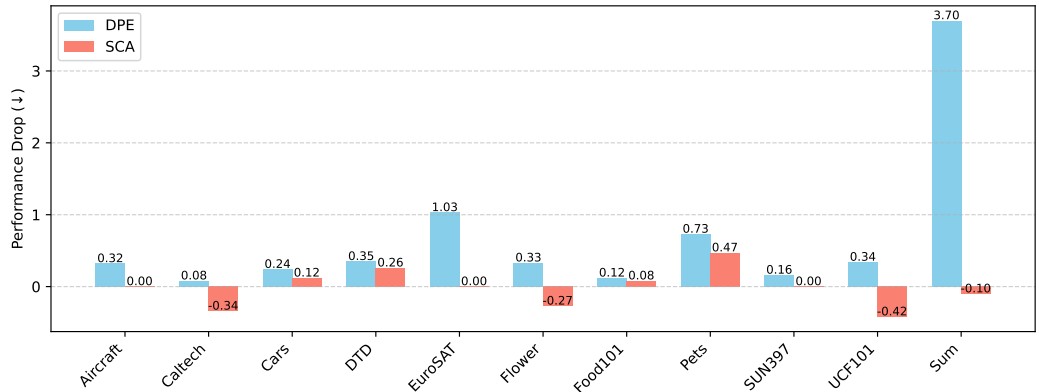

Figure A2: Forgetting issues.

overwriting the stored knowledge. (3) Remarkably, SCA even leverages these "forgetting triggers" to improve performance (a negative drop), turning a vulnerability into a strength.

**Blocking:** A misclassified sample with low entropy can, once inserted into the cache, occupy a cache slot persistently due to its high confidence score. This prevents correct but higher-entropy samples from being stored, thereby impeding subsequent knowledge accumulation and refinement. To illustrate this effect, we simulate blocking cases by randomly selecting 20% of all classes and, for each selected class, identifying the single misclassified sample with the lowest entropy. These samples are placed at the start of the test sequence, and the performance drop is measured against the standard evaluation order. As shown in the figure below, the performance degradation caused by blocking is considerably smaller for SCA than for DPE. This advantage arises because SCA's statistics-based accumulation mechanism is not constrained by fixed cache slots, thus avoiding the persistence of early high-confidence errors and enabling effective learning from later samples.

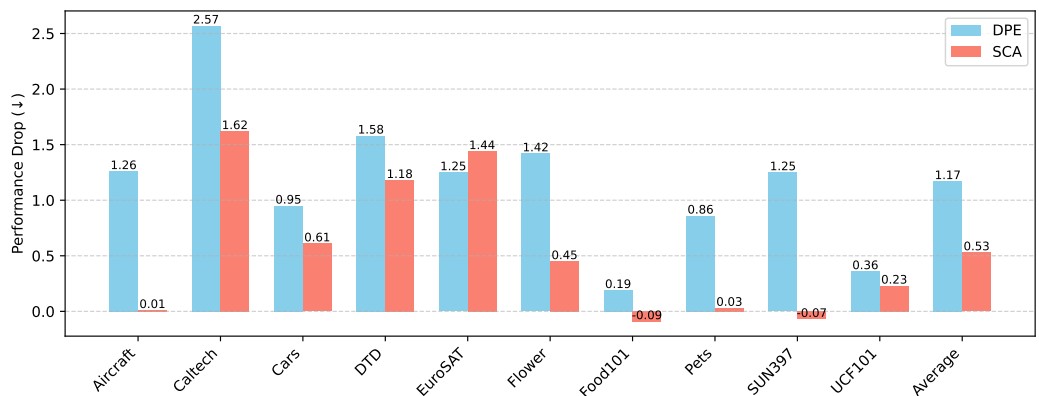

Figure A3: Blocking issues.

# B Experiment

## B.1 Details on Dataset

We evaluate our methods on 15 datasets, including Caltech101 [29], DTD [31], EuroSAT [32], FGV-CAircraft [28], Flowers102 [22], Food101 [33], OxfordPets [34], StanfordCars [30], SUN397 [35], UCF101 [36], ImageNet [37], ImageNet-V2 [10], ImageNet-Sketch [39], ImageNet-A [9], and ImageNet-R [38]. Table B1 provides detailed statistics for each dataset, including the number of

classes, the sizes of the training, validation, and testing sets, as well as the corresponding original task for each dataset.

Table B1: **Summary of the 15 image classification datasets used in experiments**. The last four ImageNet variant datasets are designed for evaluation only and contain no training or validation splits.

| Benchmark | Dataset | Classes | Splits | | | Task |
|---|---|---|---|---|---|---|
| | | | *train* | *val* | *test* | |
| Cross-Domain | Caltech101 [29] | 100 | 4,128 | 1,649 | 2,465 | Object recognition |
| | DTD [31] | 47 | 2,820 | 1,128 | 1,692 | Texture recognition |
| | EuroSAT [32] | 10 | 13,500 | 5,400 | 8,100 | Satellite imagery |
| | FGVCAircraft [28] | 100 | 3,334 | 3,333 | 3,333 | Fine-grained aircraft recognition |
| | Flowers102 [22] | 102 | 4,093 | 1,633 | 2,463 | Fine-grained flower recognition |
| | Food101 [33] | 101 | 50,500 | 20,200 | 30,300 | Fine-grained food recognition |
| | OxfordPets [34] | 37 | 2,944 | 736 | 3,669 | Fine-grained pet recognition |
| | StanfordCars [30] | 196 | 6,509 | 1,635 | 8,041 | Fine-grained car recognition |
| | SUN397 [35] | 397 | 15,880 | 3,970 | 19,850 | Scene recognition |
| | UCF101 [36] | 101 | 7,639 | 1,898 | 3,783 | Action recognition |
| Out-of-Distribution | ImageNet [37] | 1,000 | 1.28M | - | 50,000 | Object recognition |
| | ImageNet-V2 [10] | 1,000 | - | - | 10,000 | Robustness (collocation shift) |
| | ImageNet-Sketch [39] | 1,000 | - | - | 50,889 | Robustness (sketch domain) |
| | ImageNet-A [9] | 200 | - | - | 7,500 | Robustness (adversarial attack) |
| | ImageNet-R [38] | 200 | - | - | 30,000 | Robustness (multi-domain) |

## B.2  Baseline Details

- TPT [12], a prompt tuning method designed to minimize self-entropy across predictions from multiple augmented views.

- DiffTPT [13], an improved variant of TPT that leverages diffusion-based augmentations to refine prompt optimization.

- TDA [14], a cache-based method that builds positive and negative caches to improve the prediction of the current test sample.

- DMN-ZS [20], a cache-based method that constructs memory banks from test data and pseudo-labels to enhance the prediction of the current test sample.

- DPE [15], a cache-based method that trains multi-modal prototypes with historical test data to accumulate task-specific knowledge for test-time adaptation.

- BCA [27], a method that not only updates class embeddings to adapt the likelihood but also updates each class's prior using the posterior of incoming samples.

- Dynaprompt [23], a prompt-tuning method that maintains a prompt buffer and introduces a dynamic selection strategy to adaptively leverage relevant information from previous test samples for test-time adaptation.

## B.3  Comparison with other baselines

DMN-ZS [20] is a cache-based method that builds memory banks using test data and pseudo-labels to boost the prediction accuracy of the current test sample. In the original paper, DMN-ZS [20] uses a large cache size ($M = 50$) and conducts extensive hyperparameter search with ground-truth labels from the test set to determine the fusion hyperparameter for each downstream task. Removing hyperparameter search leads to a significant drop in performance. We provide a version that uses the same small cache size ($M = 3$) as other cache-based methods and applies a fixed fusion hyperparameter across all datasets. As shown in Table B2, our SCA substantially outperforms DMN-ZS under a fixed fusion hyperparameter and a small cache size ($M = 3$), and exceeds the performance of DMN-ZS configured with a large cache and per-dataset hyperparameter tuning as reported in the original paper.

Additionally, we provide results of other two gradient-based baselines WATT [50] and TENT [51] on the cross-domain benchmark using CLIP ViT-B/16.

Table B2: **Experimental results on the cross-domain benchmark with CLIP ViT-B/16.** DMN-ZS ($M$=50) refers to the original results, where the fusion hyperparameter is exhaustively tuned for each dataset. DMN-ZS ($M$=3, fixed) uses the same small cache size as other cache-based methods and applies a fixed fusion hyperparameter across all datasets. The results of TENT and WATT are directly derived from [52].

| Method | Aircraft | Caltech | Cars | DTD | EuroSAT | Flower | Food101 | Pets | SUN397 | UCF101 | Average |
|---|---|---|---|---|---|---|---|---|---|---|---|
| DMN-ZS ($M$=50) | **30.03** | **95.38** | 67.96 | 55.85 | **59.43** | 74.49 | 85.08 | **92.04** | 70.18 | 72.51 | 70.30 |
| DMN-ZS ($M$=50, fixed) | 29.76 | 94.60 | 67.53 | 55.08 | 56.79 | 75.27 | 81.60 | 89.89 | 69.95 | 72.22 | 69.27 |
| DMN-ZS ($M$=3) | 28.56 | 94.77 | 66.73 | 53.84 | 56.75 | 75.49 | 84.50 | 91.03 | 68.99 | 72.59 | 69.33 |
| DMN-ZS ($M$=3, fixed) | 25.59 | 91.52 | 55.57 | 46.63 | 46.14 | 74.83 | 71.57 | 72.17 | 60.77 | 65.16 | 61.00 |
| TENT | 23.52 | 93.87 | 65.96 | 45.51 | 49.36 | 67.40 | 82.89 | 87.35 | 65.58 | 65.08 | 64.65 |
| WATT | 24.27 | 93.27 | 66.37 | 46.39 | 55.95 | 68.21 | 83.26 | 88.09 | 65.89 | 66.01 | 65.77 |
| SCA (Ours) | 28.50 | 94.85 | **68.49** | **57.09** | 57.16 | **76.09** | **86.09** | 91.44 | **70.27** | **73.43** | **70.34** |

## B.4 Impact of prompt initialization

We follow the existing cache-based methods that use prompt ensembling strategy [20, 15] to initialize the prompt. This practice is widely adopted in the literature to ensure a robust and fixed zero-shot classifier, which serves as the foundation for these feature-space adaptation methods.

For prompt-tuning based methods, the use of a single prompt for methods like TPT and DiffTPT is also the standard protocol, as their focus is on demonstrating adaptation of the prompt itself.

Additionally, we present results using high-quality prompts obtained through training as the enhanced prompt initialization for prompt-tuning based TTA [12]. Specifically, we use the prompts trained by MaPLe [53] as the initialization, a widely adopted approach in prompt-tuning based TTA. As shown in the table below, the performance of prompt-tuning based methods is significantly lower than that of cache-based methods.

Table B3: **Experimental results on the cross-domain benchmark with CLIP ViT-B/16.**

| Method | Aircraft | Caltech | Cars | DTD | EuroSAT | Flower | Food101 | Pets | SUN397 | UCF101 | Average |
|---|---|---|---|---|---|---|---|---|---|---|---|
| TPT | 24.78 | 94.16 | 66.87 | 47.75 | 42.44 | 68.98 | 84.67 | 87.79 | 65.50 | 68.04 | 65.10 |
| TPT+MaPLe | 24.70 | 93.59 | 66.50 | 45.87 | 47.80 | 72.37 | 86.64 | 90.72 | 67.54 | 69.19 | 66.49 |
| SCA (Ours) | 28.50 | 94.85 | **68.49** | **57.09** | 57.16 | **76.09** | **86.09** | 91.44 | **70.27** | **73.43** | **70.34** |

## B.5 Impact of the number of classes and test samples on performance

**Impact of the number of classes on performance:** The number of classes inevitably impacts performance, as this dictates the inherent difficulty of the task. However, our results show that there is no consistent inverse relationship between the performance gain of SCA and the number of classes. In other words, the effectiveness of our SCA does not systematically degrade as the number of classes increases.

**Impact of the number of test samples on performance:** The performance of any cache-based TTA method, including strong baselines like DPE and our SCA, is indeed influenced by the amount of test data available for adaptation. Here, we evaluate both SCA and DPE on subsets of the test data (10%, 50%, and 100%) for two datasets: DTD (1692 testing samples) and SUN397 (19,850 testing samples). As shown in the figure below, while both methods experience performance degradation with fewer test samples, SCA shows a smaller drop. This is because SCA leverages every encountered testing sample to incrementally update its feature statistics. Unlike existing cache-based methods that may filter or discard samples, SCA ensures that no information is wasted. By fully leveraging all encountered test samples, SCA builds a more comprehensive and accurate statistical model, resulting in a notable performance advantage under data-scarce scenarios.

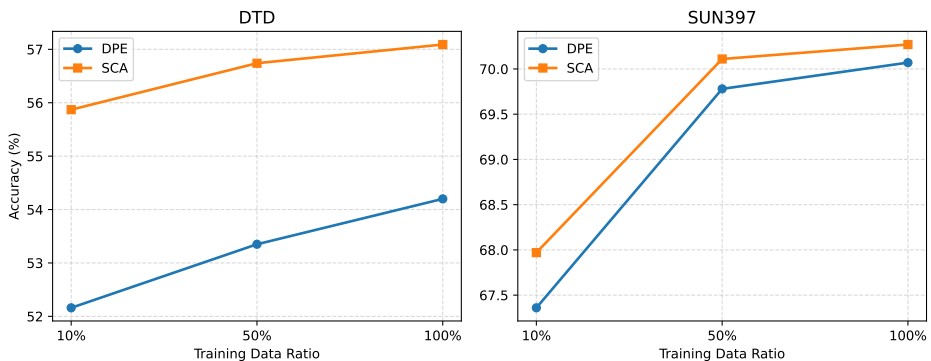

Figure B4: Performance curve.

# C  Broader Impacts

This paper focuses on test-time adaptation (TTA) for vision-language models (VLMs). In real-world applications, data often come from domains that differ significantly from the training distribution, leading to performance degradation. TTA enables VLMs to adjust to these unseen distributions on the fly, without requiring access to labeled data. Our research improves both the effectiveness and efficiency of TTA on VLMs, paving the way for its wider adoption in practical applications.

