# OpenReview forum: "Statistics Caching Test-Time Adaptation for Vision-Language Models"
_NeurIPS.cc/2025/Conference — NeurIPS 2025 poster_

### Official Review · Reviewer_JWVo · 2025-06-17

**Clarity:** 3
**Significance:** 3
**Originality:** 2
**Rating:** 5
**Confidence:** 5

**Summary:**

This paper introduces SCA, a Test-Time Adaptation strategy for VLMs (e.g. CLIP), that uses a cache of statistics. The method deviates from previous cache-based methods in that the features are not directly stored, and pseudolabels are not fully trusted based on a low entropy criterion. The pipeline is fast, as it does not require gradient propagation through the model (as in prompt tuning alternatives), and only fuses the zero-shot logits with the cache logits, using adaptive fusion parameters.

**Questions:**

1. What is the impact of adapting on a single image and not with a batch? Specially when computing the Gram matrix G, having a few more samples could introduce more information from the batch in a transductive way.
2. What would the performance be on corrupted datasets (see weaknesses). Some research ignore those datasets that were traditional in the field of TTA, and that significantly decrease the performance of CLIP.
3. The ablation study on Table 4 is not totally significant, can other datasets be included?

**Ethical Concerns:**

["NO or VERY MINOR ethics concerns only"]

**Final Justification:**

The paper proposes an interesting idea that would help push the field of gradient-free TTA for VLMs forward, mainly by desisting from keeping a memory of raw features, and instead focusing on their statistics. The results are relevant and the rebuttal answers support their initial claims. I will increase my score to 5: accept.

**Limitations:**

Yes

**Paper Formatting Concerns:**

The formatting is correct

**Quality:**

3

**Strengths And Weaknesses:**

Strengths:

1. The paper proposes an alternative to popular cache based approaches for TTA, that does not require storing features directly. The main advantage is that the criterion does not rely on low entropy, which might be deceiving; incorrect features could be stored accidentally.
2. The statistics-based cache is easy to compute, and the pseudo-label selection approach is interesting to avoid taking zero-shot predictions directly.
3. Having adaptive fusion hyperparameters is very useful, as most of previous works finetune their hyperparameters on each dataset.
4. Results show a competitive performance in several of the widely used benchmarks.

Weaknesses:

1. This work disregards another branch of TTA research that is also based on gradient, but that has proven to be competitive, such as WATT, or even TENT.
2. Other datasets that are popular in the field of TTA are not considered; corruptions (CIFAR-10/100-C, Imagenet-C), VisDA-C, etc.
3. The symbol \theta is used equally for the softmax temperature as for the cumulative probability threshold in Eq. 9.

---

> ### Author Rebuttal · Authors · 2025-07-31
>
> We thank the reviewer for taking the time to review our work. We appreciate your positive feedback on the quality and clarity of our paper. Below, we provide detailed responses to your valuable comments.
>
> ### **Q1: Impact of single and batch adaptation for accumulation**
>
> Thank you for your insightful question. We agree that adapting on a batch introduces more information that accelerate the accumulation of feature statistics. Specifically, when adapting on a single image, the accumulation of feature statistics is indeed slower, as each test step incorporates only one sample. In contrast, batch adaptation allows the model to leverage multiple samples simultaneously, enabling faster accumulation. For fair comparison, we follow the setting in previous works that compare in single adaptation (batch size = 1).
>
> ### **W2 & Q2: Performance on corrupted datasets**
>
> We thank the reviewer for this excellent suggestion to broaden our empirical evaluation. Our initial experiments focused on the challenging OOD and cross-domain benchmarks standard in recent VLM-TTA literature. However, we agree that evaluating robustness against common corruptions provides a more complete and compelling picture of our method's capabilities.
>
> Following this valuable advice, we have conducted additional experiments on **CIFAR-10-C, CIFAR-100-C, ImageNet-C, and VisDA-C**, using the same CLIP ViT-B/16 backbone. We compare our method, SCA, against the strong state-of-the-art baseline, DPE.
>
> | Method         | CIFAR10-C | CIFAR100-C | ImageNet-C |  VisDA-C  |
> | -------------- | :-------: | :--------: | :--------: | :-------: |
> | CLIP           |   60.14   |   35.16    |   24.47    |   86.78   |
> | DPE            |   62.17   |   36.48    |   24.81    |   86.43   |
> | **SCA (Ours)** | **62.84** | **36.72**  | **25.64**  | **86.95** |
>
> As shown in table,  SCA is effective against both domain shifts and common corruptions, broadening its applicability.
> ### **Q3: Ablation study on other datasets**
>
> Thank you for this suggestion. In our paper, our ablation studies were primarily conducted on the cross-domain benchmark for faster validation. Here, we provide the results of ablation study on large scale ImageNet:
>
> | Method Configuration               | ImageNet Accuracy |
> | :--------------------------------- | :---------------: |
> | w/o statistics accumulation (M=4)  |       70.79       |
> | w/o statistics accumulation (M=8)  |       70.81       |
> | w/o statistics accumulation (M=16) |       71.02       |
> | w/o statistics accumulation (M=32) |       71.08       |
> | w/o dynamic soft label assignment  |       70.65       |
> | **SCA (with both)**                |     **71.75**     |
>
> As shown in table, we have several observations: (1) our full method (`71.75%`) consistently outperforms all variants of the feature caching baseline (`w/o statistics accumulation`). This demonstrates that our approach of accumulating knowledge from all samples into compact statistics is fundamentally more effective than maintaining a limited-size cache of discrete features, even on large-scale datasets. (2) Dynamic soft label assignment plays a great role in performance gain. This underscores the critical importance of this component, especially on a challenging, large-scale dataset with 1000 classes where the risk of generating incorrect hard pseudo-labels is extremely high. This confirms that our uncertainty-aware labeling is essential for robust adaptation.
>
> ### **W1: Comparison with WATT and TENT**
>
> We prvoid results of WATT and TENT on cross-domain benchmark using CLIP ViT-B/16. We will prvoide results of WATT and TENT using different backbone and datasets in our final version.
>
>
> |                | Aircraft | Caltech | Cars  |  DTD  | EuroSAT | Flower | Food101 | Pets  | SUN397 | UCF101 | Average |
> | -------------- | :------: | :-----: | :---: | :---: | :-----: | :----: | :-----: | :---: | :----: | :----: | :-----: |
> | CLIP-ViT-B/16  |  23.67   |  93.35  | 65.48 | 44.27 |  42.01  | 67.44  |  83.65  | 88.25 | 62.59  | 65.13  |  63.58  |
> | TENT           |  23.52   |  93.87  | 65.96 | 45.51 |  49.36  | 67.40  |  82.89  | 87.35 | 65.58  | 65.08  |  64.65  |
> | WATT           |  24.27   |  93.27  | 66.37 | 46.39 |  55.95  | 68.21  |  83.26  | 88.09 | 65.89  | 66.01  |  65.77  |
> | DPE            |  28.95   |  94.81  | 67.31 | 54.20 |  55.79  | 75.07  |  86.17  | 91.14 | 70.07  | 70.44  |  69.40  |
> | **SCA (Ours)** |  28.50   |  94.85  | 68.49 | 57.09 |  57.16  | 76.09  |  86.09  | 91.44 | 70.27  | 73.43  |  70.34  |
>
>
> ### **W3: Ambiguous notation**
>
> Thank you for pointing this out. We will fix it in the final version for clarity.
>
> Thank you for your thoughtful review. If there are any concerns or if further clarification is needed, we are more than happy to provide additional details or revise accordingly.

---

> > ### Comment · Reviewer_JWVo · 2025-08-02
> > **Follow up discussion**
> >
> > Thank you for answering to my questions. I would like to add the following:
> >
> > 1. In your experiments with new datasets (__Q2__), the improvement doesn't damp when the number of classes grow. In fact, Imagenet-C shows one of the largest gains among the four datasets. Is this an indicator that SCA is not sensitive to the number of classes? On the contrary, it looks like it can be more limited when tested on smaller datasets, as less statistics are available.
> >
> > 2. SCA combines the original model's logits and the cache logits. Based on your analysis, do the cache logits contribute in boosting performance or in correcting the model's logits when they are less accurate?

---

> > > ### Author Response · Authors · 2025-08-03
> > >
> > > Thank you for your timely feedback. We appreciate your positive attitude toward our work and your insightful questions. Here, we provide detailed responses:
> > >
> > > ### **Q1: Impact of the number of classes and test samples on performance**
> > >
> > > **Impact of the number of classes on performance:** The number of classes inevitably impacts performance, as this dictates the inherent difficulty of the task. However, as you observed, our results show that there is no **consistent inverse** relationship between the performance gain of SCA and the number of classes. In other words, the effectiveness of our SCA does not systematically degrade as the number of classes increases.
> > >
> > > **Impact of the number of test samples on performance:** The performance of any cache-based TTA method, including strong baselines like DPE and our SCA, is indeed influenced by the amount of test data available for adaptation. Here, we evaluate both SCA and DPE on subsets of the test data (10%, 50%, and 100%) for two datasets: **DTD** (1692 testing samples) and **SUN397** (19,850 testing samples). The results are presented in the table below:
> > >
> > > |     | DTD (10%) | DTD (50%) | DTD (100%) | SUN397 (10%) | SUN397 (50%) | SUN397 (100%) |
> > > | :-- | :-------: | :-------: | :--------: | :----------: | :----------: | :-----------: |
> > > | DPE |   52.16   |   53.35   |   54.20    |    67.36     |    69.78     |     70.07     |
> > > | SCA | **55.87** | **56.74** | **57.09**  |  **67.97**   |  **70.11**   |   **70.27**   |
> > >
> > > As shown in the table above, while both methods experience performance degradation with fewer test samples, SCA shows a smaller drop. This is because **SCA leverages every encountered testing sample to incrementally update its feature statistics**. Unlike existing cache-based methods that may filter or discard samples, SCA ensures that no information is wasted. By fully leveraging all encountered test samples, SCA builds a more comprehensive and accurate statistical model, resulting in a notable performance advantage under data-scarce scenarios.
> > >
> > > ### **Q2: The role of the cache logits**
> > >
> > > The ultimate purpose of the cache logits is to boost the overall test-time adaptation performance. SCA achieves this by adaptively fusing the cache logits (capture domain-specific knowledge from the test samples) with the original model's zero-shot logits (encapsulate general and pre-trained knowledge). The specific role of cache logits is not fixed but is dynamically determined for each sample based on the original model's confidence:
> > > * When the original model is uncertain about its prediction for a given sample, the adaptive fusion mechanism assigns a relative large weight to the cache logits. The model reduces its reliance on the uncertain zero-shot logits and instead rely primarily on the cache logits to guide the final prediction.
> > > * When the original model is confident about its prediction for a given sample, the fusion mechanism assigns a relative small weight to the cache logits. In this case, the cache logits may either reinforce the original prediction to enhance its stability and decision margin, or provide a counter-signal that helps correct a potential high-confidence error.
> > >
> > >
> > > Thank you once again for your valuable feedback. If you have any additional concerns or comments that we may have missed in our responses, we would be most grateful for any further feedback from you to help us further enhance our work.

---

> > > > ### Comment · Reviewer_JWVo · 2025-08-05
> > > > **Follow up**
> > > >
> > > > Thanks to the authors for engaging in the discussion and clarifying my questions. in the light of the quality of the rebuttal and the overall contribution of the paper, I will increase my score one unit. I look forward to see the important discussions suggested by the other reviewers and myself in a final version of this paper.

---

> > > > > ### Author Response · Authors · 2025-08-05
> > > > >
> > > > > Thank you sincerely for your thoughtful feedback and the updated score. We deeply appreciate your support and will carefully incorporate your suggestions and those from the other reviewers in the final version.

---

### Official Review · Reviewer_EE62 · 2025-06-30

**Clarity:** 3
**Significance:** 2
**Originality:** 3
**Rating:** 4
**Confidence:** 4

**Summary:**

This work focuses on test-time adaptation for vision-language models. The authors claim that existing methods struggle to achieve robust and continuous knowledge accumulation during test time and propose Statistics Caching test-time Adaptation (SCA), a novel cache-based approach. Specifically, SCA formulates the reuse of past features as a least squares problem, avoiding the storage of raw features and instead maintaining compact representations. In addition, SCA introduces adaptive strategies to reduce the impact of noisy pseudo-labels and dynamically balance multiple prediction sources. Extensive experiments on two benchmarks demonstrate that SCA achieves strong performance.

**Questions:**

Please see the weakness.

**Ethical Concerns:**

["NO or VERY MINOR ethics concerns only"]

**Final Justification:**

My concern has been addressed. Overall, I am inclined to give a borderline accept.

**Limitations:**

yes

**Quality:**

3

**Strengths And Weaknesses:**

**Strength**

1. The paper proposes Statistics Caching test-time Adaptation (SCA), a novel cache-based approach that enables robust and continuous knowledge accumulation during test time. Storing compact information instead of raw features is a promising alternative to enable efficient adaptation.

2. Comprehensive experiments, including comparisons with state-of-the-art methods and ablation studies, are provided.

**Weakness**

1. The motivation requires more detailed explanation. Although the authors provide Figure 2(a) to illustrate that exsiting approach leads to performance degradation, it would be helpful to analyze what types of features are forgotten during adaptation and how knowledge forgetting affects performance. In addition, how do low-entropy and high-entropy samples affect the learning process and contribute to information forgetting? The authors could also provide performance curves of the proposed method as a comparison to better demonstrate its effectiveness.

2. The authors formulate feature storage as a least squares problem. It would be better if they could provide some theoretical justification or insights to explain why this approach yields better compact features for test samples. Also, the authors could compare it with other feature compaction strategies, such as using class prototypes and updating them with certain rules.

3. Prediction fusion is a common strategy in VLM adaptation for improving prediction quality. It is somewhat conflicting to use entropy to weight predictions from different sources, especially since the authors themselves note that entropy is not a reliable measure of prediction accuracy (e.g., lines 40–69). Have the authors tried alternative weighting strategies, and have they analyzed the robustness of the entropy-based weighting in assigning correct importance to different predictions?

4. I noticed that the authors use a prompt ensembling strategy to initialize prompts. However, prior work such as TPT and DiffTPT uses only a single prompt ("a photo of a") for initialization. This may introduce unfairness in comparison, as the ensemble strategy can yield a stronger baseline.

5. Some feature visualizations should be provided, as they would help illustrate how stored features contribute to adaptation.

---

> ### Author Rebuttal · Authors · 2025-07-31
>
> We would like to thank the reviewer for taking the time to review our work. We appreciate your recognition of the originality of our paper. According to your valuable comments, we provide detailed feedback.
>
> ### **W1: Motivation**
>
> Thank you for this constructive feedback. *Firstly, we briefly introduce the mechanism of existing cache-based methods.*
>
> Existing cache-based methods, like TDA, maintain a fixed-size feature cache for each class during test time. The process works as follows:
>
> **1. Cache Update Mechanism:**
> For each incoming test sample, the model first generates a pseudo-label (predicted class) and a corresponding confidence score (**typically measured by prediction entropy**). It then attempts to update the cache associated **with the predicted class**:
>
> *   The model locates the specific cache for the predicted class.
> *   If the cache for the predicted class is not full: The sample's feature is added to it.
> *   If the cache for the predicted class is full: The new sample's confidence is compared to the confidence of the features already stored in it. If the new sample is **more confident** than the **least confident feature** in the cache for predicted class, its feature **replaces** that one. Otherwise, the new sample is discarded.
>
> **2. Prediction:**
> The final prediction for a new sample is a combination of the model's original zero-shot prediction and a new prediction derived from comparing the sample's feature similarity to the features stored across all class-specific caches.
>
> *Then we provoide a analysis to address your specific points:*
>
> **What types of features are forgotten during adaptation.** To be precise, the forgetting issue we address in this paper refers to a correct but relatively low-confidence (high-entropy) feature being overwritten by an incorrect but high-confidence (low-entropy) one. For example, on the DTD dataset, a canonical "striped" texture feature might be overwritten by a misclassified "lined" texture that the model confidently mistakes for "striped."
>
> **How it Affects Performance:** the model's predictions rely on comparing new samples to these cached features. When a correct cached feature is replaced by an incorrect one, the model's ability to correctly match samples is impaired, leading to misclassifications and a performance drop, as shown in Figure 2(a). Our SCA method, by accumulating statistics from *all* samples rather than maintaining a fixed-size "winner-takes-all" cache, is immune to this specific type of forgetting.
>
> **The Role of Low and High-entropy samples:**
>
> * Correct low-entropy (high-confidence) samples serve as strong positive examples. They reinforce the quality of class representations in the cache, leading to more accurate future predictions.
> * Incorrect low-entropy samples may replace correct ones in the cache, causing **forgetting**. Because of their high confidence, these incorrect samples can hold cache slots for a long time, **blocking** later higher-entropy but correct samples.
> * High-entropy (low-confidence) samples typically fail to enter the cache, which leads to the loss of potentially useful, though uncertain, information. Our SCA enables these potential samples to contribute to the accumulated statistics in a controlled way through dynamic soft label assignment.
>
> In the original version, we included more details on the forgetting and blocking issues in the **Supplementary Material (Appendix A)** due to page limits. In the final version, we will move this content into the main paper and add more visualizations, such as **the performance curves of our method in Figure 2(a)**, to make the motivation clearer.
>
> ### **W2: Rationality of our accumulation scheme**
>
> Thank you for this excellent question, which allows us to elaborate on the theoretical motivation behind our approach.
>
> **1. Theoretical Justification for the Least Squares Formulation:**
>
> Our core idea is to move beyond simple feature matching and instead learn a linear classifier ($\boldsymbol {W} _{t}$) directly in the feature space, which adapts over time. The least squares formulation
>
> $$ \min_{\boldsymbol{W} _t}  \|\| \boldsymbol{X} _{1:t} \boldsymbol{W} _{t} - \boldsymbol{Y} _{1:t}  \|\| _\mathrm{F}^2 + \gamma \|\|\boldsymbol{W} _{t} \|\| _\mathrm{F}^2$$
>
> is the classic way to achieve this for classification. The closed-form soultion is
>
> $$\boldsymbol{W} _t=(\boldsymbol{X} _{1:t}^\top\boldsymbol{X} _{1:t}+\gamma\boldsymbol{I})^{-1}\boldsymbol{X} _{1:t}^\top\boldsymbol{Y} _{1:t}.$$
>
> As this solution shows, to compute the optimal classifier $\boldsymbol {W} _t$, we **do not need to store the entire, ever-growing feature matrix $\boldsymbol{X} _{1:t}\in \mathbb{R}^{n _{1:t} \times d}$** ($n _{1:t}$ denotes the number of features extracted from all test samples up to time $t$, $d$ denotes the feature dimension). We only need store two **fixed size** statistics :
>
> *   $\boldsymbol{G} _{1:t}=\boldsymbol{X} _{1:t}^\top \boldsymbol{X} _{1:t}=\boldsymbol{G} _{1:t-1}+\boldsymbol{X} _t^\top\boldsymbol{X}_t$ (the Gram matrix): This captures the **second-order statistics** of the features. It encodes the covariance structure between all feature dimensions.
> *   $\boldsymbol{C} _{1:t}=\boldsymbol{X} _{1:t}^\top \boldsymbol{Y} _{1:t}=\boldsymbol{C} _{1:t-1}+\boldsymbol{X} _t^\top\boldsymbol{Y} _t$: This captures the **first-order statistics** relating features to their class labels.
>
> Therefore, formulating the problem as least squares naturally leads to a highly compact and efficient online learning scheme. We are not arbitrarily "compacting features." Instead, we are storing the essential sufficient statistics necessary to train a robust linear classifier using all the data encountered so far. This provides a strong theoretical grounding for our statistics caching approach.
>
> **2. Comparison with Class Prototype Methods:**
>
> Class prototype methods only store and update the mean feature vector for each class. This is a **first-order statistics** approach. It implicitly assumes that classes form simple, spherical clusters in the feature space and only captures the *center* of each class. It's conceptually similar to a nearest-neighbor classifier.
>
> Our SCA additionally uses **second-order statistics**. This means our model understands not just the *center* of each class's data cloud, but also its shape, orientation, and variance. This allows the learned classifier to create much more sophisticated decision boundaries.  We also provide the results of class prototypes methods on cross-domain benchmark using ViT-B/16:
>
> ||Avg.|
> |-|:-:|
> |Class Prototype|65.23|
> |SCA|70.34|
>
> As shown in the table above, our method significantly outperforms Class Prototype methods.
> ### **W3: Robustness of weighting strategies for prediction fusion**
>
> Thanks for your suggetsion. We provide the average accuracy of our SCA on cross-domain benchmark using different weighting strategies, including softmax probability [1] and mahalanobis distance [2] :
>
> ||Entropy (Default) |Softmax|Mahalanobis distance|
> |-|-|-|-|
> |SCA|70.34|70.27|70.22|
>
> As shown in the table, our SCA method is compatible with various weighting strategies and show robust performance. We selected entropy due to its simplicity and widespread adoption in the literature.
>
> [1] A Baseline for Detecting Misclassified and Out-of-Distribution Examples in Neural Networks, ICLR'17\
> [2] A Simple Unified Framework for Detecting Out-of-Distribution Samples and Adversarial Attacks, NeurIPS'18.
>
> ### **W4: Prompt Initialization**
>
> We follow the existing cache-based methods that use prompt ensembling strategy to initialize the prompt. This practice is widely adopted in the literature to ensure a robust and fixed zero-shot classifier, which serves as the foundation for these feature-space adaptation methods.
>
> For prompt-tuning based methods, the use of a single prompt for methods like TPT and DiffTPT is also the standard protocol, as their focus is on **demonstrating adaptation of the prompt itself.**
>
> Additionally, we present results using high-quality prompts obtained through training as the enhanced prompt initialization for prompt-tuning based TTA [1]. Specifically, we use the prompts trained by MaPLe [2] as the initialization, a widely adopted approach in prompt-tuning based TTA.  As shown in the table below, the performance of prompt-tuning based methods is significantly lower than that of cache-based methods.
>
> | | Aircraft | Caltech | Cars  |  DTD  | EuroSAT | Flower | Food101 | Pets  | SUN397 | UCF101 | Average |
> | - | :------: | :-----: | :---: | :---: | :-----: | :----: | :-----: | :---: | :----: | :----: | :-------: |
> | TPT |  24.78|  94.16 | 66.87 | 47.75 |  42.44  | 68.98  |  84.67  | 87.79 | 65.50  | 68.04  | 65.10   |
> | TPT+MaPLe |  24.70   |  93.59  | 66.50 | 45.87 |  47.80  | 72.37  |  86.64  | 90.72 | 67.54  | 69.19  | 66.49   |
> | DPE |  28.95 |  94.81  | 67.31 | 54.20 |  55.79  | 75.07  |  86.17  | 91.14 | 70.07  | 70.44  | 69.40   |
> | Ours  |  28.50   |  94.85  | 68.49 | 57.09 |  57.16  | 76.09  |  86.09  | 91.44 | 70.27  | 73.43  | 70.34 |
>
> [1] Test-Time Prompt Tuning for Zero-Shot Generalization in Vision-Language Models, NeurIPS'22\
> [2] Maple: Multi-modal prompt learning, CVPR'23
>
> ### **W5: Feature visualizations**
>
> Thank you for this suggestion. Existing cache-based methods store the features of specific samples, allowing for the use of visualization techniques, such as t-SNE, to directly assess cache quality. In contrast, our SCA method does not store the features themselves but rather feature statistics that summarize information from all past samples, making such visualization methods less applicable. In the final version, we will include performance curve visualizations to compare the impact of the features or feature statistics stored by different methods on adaptation.
>
> Thank you for your thoughtful review. If there are any concerns or if further clarification is needed, we are more than happy to provide additional details accordingly.

---

> > ### Comment · Reviewer_EE62 · 2025-08-04
> >
> > Thanks for the response. I still have some concerns about using entropy as the fusion weight, especially since the paper itself notes that entropy is not a reliable measure of prediction accuracy. Although different strategies seem to give similar results, the main focus of this work is the problem of entropy, which makes this choice more critical.

---

> > > ### Author Response · Authors · 2025-08-05
> > >
> > > We sincerely appreciate your valuable feedback. Here, we provide detailed responses:
> > >
> > > Entropy is an effective metric for estimating prediction confidence. While confidence is not a perfect proxy for prediction accuracy, **this does not mean we should disregard it entirely**. In test-time adaptation (TTA), where true labels aren't available, confidence is one of the few practical signals we can rely on (entropy is the most commonly used confidence measure). That's why **most existing TTA methods are still built around entropy (e.g. entropy minimization).**
> > >
> > > Besides, **the main focus of this work is not the problem of entropy, but on how it is used.** Existing cache-based methods rely entirely on entropy to determine which samples enter or leave the cache, implicitly assuming that high-confidence (low-entropy) predictions are always correct. This strategy can lead to the forgetting and blocking issues we highlight in the introduction, as overconfident incorrect predictions may misguide cache updates.
> > >
> > > In contrast, our Instance-level Adaptive Fusion strategy **does not blindly trust high-confidence predictions**. Instead, we compare the entropy of two different classifiers (the original zero-shot model and our statistics-based cache model) on the exact same test sample. This allows the model to assess their **relative confidence**, relying more on the zero-shot model when the cache model is uncertain, and shifting toward the cache model as it becomes more confident. Several works also use this confidence-based fusion with various confidence metrics in other machine learning area [1,2,3]. We use entropy here because it is the most common confidence metric. As noted in our previous rebuttal, replacing it with other confidence metrics yields similar results.
> > >
> > > Finally, we would like to emphasize that the another primary role of this adaptive fusion mechanism is to **avoid costly, dataset-specific hyperparameter tuning** (Lines 190-194, 286-293). By automatically determining fusion weights for each sample, our method achieves robust and generalizable performance across multiple datasets. While a carefully tuned fusion coefficient **for each individual dataset**, as done in methods like DPE, may yield slightly better performance, our approach offers a strong trade-off between accuracy and the cost of hyperparameter tuning.
> > >
> > > [1] AMU-Tuning: Effective Logit Bias for CLIP-based Few-shot Learning, CVPR'24\
> > > [2] Uncertainty-Aware Fusion: An Ensemble Framework for Mitigating Hallucinations in Large Language Models, WWW'25\
> > > [3] Vision-Language Model Selection and Reuse for Downstream Adaptation, ICML'25
> > >
> > > Thank you again for your valuable feedback. If there are any concerns or comments we overlooked in our responses, we would greatly appreciate any further input to help us improve our work.

---

> > > > ### Comment · Reviewer_EE62 · 2025-08-06
> > > >
> > > > Thanks for the further clarification. I think this discussion is valuable, and the authors may consider adding it to the revised paper. I will increase my score.

---

> > > > > ### Author Response · Authors · 2025-08-06
> > > > >
> > > > > Thank you sincerely for your thoughtful feedback and the updated score. We deeply appreciate your support and will carefully incorporate this discussion in the final version.

---

### Official Review · Reviewer_mpEP · 2025-06-30

**Clarity:** 1
**Significance:** 2
**Originality:** 3
**Rating:** 4
**Confidence:** 4

**Summary:**

This paper investigates test-time adaptation (TTA) for vision-language models (VLMs) such as CLIP. In this field, two approaches have been proposed: (i) test-time prompt tuning, which optimizes trainable prompt parameters, and (ii) feature caching, which preserves past features and uses them for inference on new test samples. Test-time prompt tuning is computationally expensive because it optimizes prompts using backpropagation with unsupervised objectives such as entropy minimization. On the other hand, feature caching is computationally efficient because it does not perform optimization and instead uses past effective features classified with pseudo labels for inference. However, the paper argues that existing feature caching suffers from "forgetting" and "blocking" due to misclassified sample features because the cache size is limited. To address this, the paper proposes statistics caching test-time adaptation (SCA) to extract useful knowledge from all past test sample features. SCA formulates the classification problem using past features, such as ridge regression, and sequentially updates the Gram matrix required to obtain the closed-form solution. Additionally, to handle noisy pseudo-labels, it introduces dynamic soft pseudo-label assignment, which uses only the class with the highest prediction probability. SCA constructs the final prediction as the entropy-weighted sum of the zero-shot logits from CLIP and the logits computed from the cache. Experiments demonstrate that SCA achieves superior performance and high computational efficiency compared to recent TTA methods for VLMs, including test-time prompt tuning and feature caching.

**Questions:**

Please see the weakness section and address the concerns.

**Ethical Concerns:**

["NO or VERY MINOR ethics concerns only"]

**Final Justification:**

The rebuttal and follow-up discussions have largely addressed my concerns with concrete evidence. Therefore, I recommend that the paper be accepted. Please see our discussions for more details.

**Limitations:**

The main paper does not include a limitation section. The limitation should be discussed in the main paper.

**Paper Formatting Concerns:**

Nothing to report.

**Quality:**

2

**Strengths And Weaknesses:**

### Strengths
+ **S1.** The proposed method in this paper significantly reduces storage costs compared to existing methods by representing information from all past test samples as a Gram matrix. Leveraging all of the test samples for TTA might be a new perspective.
+ **S2.** The dynamic soft pseudo-label assignment proposed in this paper has been successful in significantly improving performance.
### Weaknesses
- **W1.** The forgetting and blocking issues arising from existing feature caching, which are the primary motivations for this paper, may not be practically significant challenges. First, as shown in Figure 2, these issues have only been reported in a limited number of datasets, indicating limited generalizability. Second, as shown in Table 4, the performance improvements of the proposed method are primarily attributed to dynamic soft label assignment. Without dynamic soft label assignment, the average DPE and SCA performance would be comparable (69.40 in Table 1 vs. 69.32 in Table 4). Since this technique is not directly and necessarily related to feature caching, it can be combined with DPE. Addressing these concerns would further strengthen the clarity of the paper's contributions.
- **W2.** The negative aspects that may be introduced by the proposed method have not been sufficiently discussed. The proposed method sequentially accumulates knowledge from all test samples into a Gram matrix. As the paper claims, this certainly has positive effects, but it may also have negative effects. For example, in situations where multiple domain shifts coexist, if samples belonging to a specific domain shift become dominant, the Gram matrix will contain statistics biased toward that domain shift. In such a situation, if samples with minor domain shifts appear, can the proposed method robustly handle them? The paper raises a primary research question, “Is this acclaimed knowledge accumulation process truly robust and effective over time?” on L42, but the evidence that the proposed method addresses this question is insufficient.
- **W3.** The method for determining hyperparameters is unclear. The paper introduces three hyperparameters, $\gamma, \beta, \tau$, into the proposed method and states that these are fixed to specific values in the experiments (L211-212). How were these fixed values discovered? As a result, the sensitivity of $\beta$ and $\gamma$ is confirmed in Figure 5, but nothing is mentioned about $\gamma$. If the tuning was based on the average value of the test set, this would damage the reliability of the paper. In any case, the protocol for determining the hyperparameters should be clearly stated.

---

> ### Author Rebuttal · Authors · 2025-07-31
>
> We sincerely thank the reviewer for their constructive and insightful feedback. The comments help us to significantly improve the clarity and rigor of our paper. We address the concerns below.
>
> ### **W1 (part 1): On the significance of the forgetting and blocking issues**
>
> Thank you for raising this important point about the significance of forgetting and blocking issues. We clarify that these are indeed practically significant because they stem from a **fundamental algorithmic vulnerability**.
>
> This vulnerability is inherent in any cache-based method that combines a fixed-size cache with a confidence-based replacement strategy. For any given dataset, a population of both low-entropy, incorrect samples and high-entropy, correct samples typicall always exist. Every time a low-entropy, incorrect sample arrives, there is a chance it will displace a correct feature from the cache, resulting in **forgetting**. And when a highly confident incorrect sample arrives early, **blocking** issues occurs.
>
> In a real-world online setting, we cannot control or guarantee the arrival order of these samples. This makes the system's performance highly susceptible to "unlucky" sequences that trigger failure modes. Much like a security flaw in software, this design represents a **persistent threat.** While it may not be exploited in every single instance, its potential to cause performance degradation when triggered.
>
> ### **W1 (part 2): On the synergy between statistics accumulation and dynamic soft labeling**
>
> Thank you for this insightful question. We respectfully argue that our dynamic soft labeling technique is **not a simple "plug-and-play" module** that can be easily combined with DPE for similar gains. DPE, like other cache-based methods, maintains a cache of **discrete, raw feature vectors** in distinct, class-specific slots. This **discrete** storage design is ill-suited for a soft pseudo-label like `{'cat': 0.7, 'dog': 0.3}`, as it raises following challenges:
> *   **How should the feature be stored?** Should it be placed in the 'cat' cache with a weight of 0.7 and the 'dog' cache with a weight of 0.3? This would require storing an additional confidence weight with each feature, complicating the cache structure.
> *   **How would this affect cache occupancy?** If one sample now occupies slots in multiple class caches, does it count towards the size limit $M$ for each of them? This could lead to a single, highly uncertain sample disproportionately blocking space for many other more confident samples.
> *   **How would the replacement mechanism work?** When considering replacing a feature in the 'cat' cache, should its confidence be its standalone prediction confidence, or should it be discounted by its soft label weight (0.7)? If a new, confident 'cat' sample arrives, should it be allowed to replace the 'cat' portion of our soft-labeled sample? Doing so would break the integrity of the original soft label, leaving a partial entry in the 'dog' cache that no longer reflects the full uncertainty of the original prediction.
>
> In contrast, in our SCA framework, dynamic soft label assignment is highly compatible with statistics accumulation. More specifically, they form a synergistic relationship:
>
> * **Statistics accumulation *unlocks the full potential* of soft labeling:** The update rule for our first-order statistics, $\boldsymbol{C} _{1:t-1}+\boldsymbol{X} _t^\top\boldsymbol{Y} _t$ naturally handles a soft probability vector $\boldsymbol{Y} _t$. The contribution of each feature to the class correlations is weighted by its soft probability, which is mathematically elegant and effective.
> * **Soft labeling *enables* robust statistics accumulation:** Accumulating knowledge from every single sample is powerful but risky. If we were to use hard labels, errors would quickly accumulate and corrupt our statistics. Our dynamic soft labeling strategy acts as an uncertainty-aware regularizer to make our statistics accumulation strategy robust.
>
> In conclusion, our dynamic soft labeling is not a simple plug-and-play module. It is tightly integrated with our statistics accumulation framework, with each part supporting the other. This synergy is key to our strong performance.
>
> ### **W2: Negative aspects of accumulation in multiple domain settings**
>
> Thank you for your insightful comment. Consistent with existing CLIP TTA methods, our approach primarily focuses on the single-domain setting, where test data come from a single dataset. Extending CLIP TTA methods to the multi-domain setting, where test samples from different domains are mixed, is a valuable and important direction. However, to the best of our knowledge, there are currently no TTA methods for VLMs that explicitly claim to address this scenario.
>
> Indeed, in the multi-domain setting, accumulation in our scheme may introduce negative effects. As a potential solution, we note that feature representations from different domains often exhibit some degree of difference. This property can be used to identify a sample's domain and accumulate feature statistics separately for each domain. We consider this a promising direction for future work and will add a discussion of this limitation and potential extensions in the final version.
>
> ### **W3: Hyperparameters selection**
>
> We select hyperparameters based on the validation set of DTD, a relatively challenging dataset (with CLIP ViT zero-shot accuracy of 44.27). Additionally, DTD has a small number of test samples, which allows for efficient validation. That's why DTD appears frequently across multiple tables.
>
> The main text omits the hyperparameter analysis of the ridge coefficient $\gamma$. Here, we present results on a cross-domain benchmark using ViT-B/16 by varying $\gamma$ from 0 to 1e6. As shown in the table below, setting $\gamma$ to zero leads to numerical instability due to a singular Gram matrix. Within the range from 1e2 to 1e6, performance remains stable, indicating that any value within this interval is appropriate.
>
> |         |  0   |   1   |  10   |  100  |  1e3  |  1e4  |  1e5  |  1e6  |
> | ------- | :--: | :---: | :---: | :---: | :---: | :---: | :---: | :---: |
> | Avg Acc | None | 70.22 | 70.24 | 70.31 | 70.31 | 70.34 | 70.35 | 70.30 |
>
> ### **L1: limitation section**
>
> Thanks for your suggestion. We will move it from suplementary material to the main text in the final version.
>
> We greatly appreciate your insightful review and are open to discussing any remaining issues. Please feel free to contact us if additional changes or clarifications are needed.

---

> > ### Comment · Reviewer_mpEP · 2025-08-04
> >
> > I would like to thank the authors for their detailed response. I would appreciate it if the authors could reconsider the remaining concerns below.
> >
> > # W1 (part 1)
> > I believe that concrete evidence is still lacking to support the claim that forgetting and blocking are "fundamental algorithmic vulnerabilities." For example, the claim would be more convincing if Figure 2 showed the average results across all datasets, not just DTD and Stanford Cars. If these issues are significant in specific datasets, it is important to clarify the conditions and assumptions under which they arise.
> >
> > # W1 (part 2)
> > As the authors explain, dynamic soft pseudo-label assignment may not be completely plug-and-play with existing cache-based methods. However, this does not fully address the concern. The most important point here is that it is unclear how this technique is theoretically related to the forgetting/blocking issues that are the main focus of this paper. According to the authors' explanations in the paper and their rebuttal, using hard labels causes cumulative statistics to collapse rapidly. This would be convincing if it were based on actual observations. However, Table 4 shows that using hard labels maintains performance from the baseline. This fact may suggest that the collapse of statistics caused by the use of hard labels, as claimed in the paper, is either not occurring or is not a significant issue in the design of the method. The paper should provide theoretical and/or experimental evidence to demonstrate what changes occur beyond performance when soft labels are used, and why these changes resolve forgetting/blocking issues. The current evaluations are biased toward task accuracy and provide limited insight in this regard.

---

> > > ### Author Response · Authors · 2025-08-06
> > >
> > > ### **W1 (part 2)**
> > >
> > > We would like to clarify that forgetting & blocking issues are solved primarily by our statistics caching. Dynamic soft label assignment is mainly designed to ensure the quality and robustness of the accumulated knowledge, specifically by mitigating pseudo-label noise. When the model's top prediction is incorrect, a hard label forces us to update our first-order statistics with a 100% confident but completely false signal. We term this the **error amplification** problem. A continuous stream of such errors will inevitably skew the learned cache classifier and degrade its accuracy.
> > >
> > > You may be confused as to why using hard labels does not cause "statistics collapse," which typically means a dramatic performance drop, conflicting with the observations ("Table 4 shows that using hard labels maintains performance from the baseline"). We apologize if our words led to this misunderstanding. In both the manuscript and our rebuttal, we have been careful to use terms such as "amplify errors," "lead to error propagation," and "corrupt statistics," rather than "statistics collapse."Our claim is that **hard labels results in error amplification, which degrades performance to some extent, rather than causing an immediate catastrophic failure.**
> > >
> > > The observation is **not in conflict** with the existence of the error amplification problem. It is the dynamic soft label assignment that alleviates this issue, leading to notable performance gains and enabling our method to surpass the baseline.
> > >
> > > We sincerely thank you for your valuable feedback. If there remain any concerns or ideas not covered in our responses, please feel free to share them. Your further input will greatly aid us in strengthening our work.

---

> > > > ### Comment · Reviewer_mpEP · 2025-08-07
> > > >
> > > > I appreciate the additional responses from authors.
> > > >
> > > > ## W1 (part 1)
> > > > Thank you for the insightful response with concrete evidence. These control experiments strongly clarify the motivation of this paper, and they should be included in the Experiments section. Now, my concerns on the reality of forgetting and blocking issues are completely addressed.
> > > >
> > > > ## W1 (part 2)
> > > > This new explanation seems to be somewhat disconnected from the claims made in the submitted paper and the rebuttal, but the error amplification problem is more convincing in my opinion. Please revise the paper based on these discussions as motivation for using soft labels.
> > > >
> > > > Finally, I believe that sufficient evidence has been presented to support the claims made in the paper through the rebuttal. Including these discussions will make the paper more useful to the community. I would like to thank the authors for their scientific and persistent responses. I hope that the final version will be further strengthened. I will increase my score toward acceptance accordingly.

---

> > > > > ### Author Response · Authors · 2025-08-07
> > > > >
> > > > > We sincerely appreciate your thoughtful feedback and support, and will incorporate the corresponding revisions in the final version.

---

> ### Author Response · Authors · 2025-08-06
>
> Thank you for this insightful comment. Here, we provide the detailed response.
>
> ### **W1 (part 1)**
>
> **1.Forgetting issues**
>
> To better illustrate the forgetting issues, we conduct an additional set of control experiments.
>
> Firstly, we identify a small set of "forgetting trigger samples." For each dataset, we select 10 random classes and find the single, lowest-entropy misclassified sample for each class. To ensure these 10 trigger samples are admitted by the cache, we filter out any correctly classified samples from the entire test set that have an even lower entropy.
>
> From the remaining pool of test samples, we randomly draw 500 samples to serve as our fixed "final test samples." All other remaining samples are designated as "initial samples," used to allow the cache to accumulate sufficient class-specific knowledge. Next, we consider the following two sequences to update the model's cache:
>
> - Sequence A: \[initial samples]
> - Sequence B: \[initial samples, forgetting trigger samples]
>
> The forgetting trigger samples, with their low entropy, are designed to infiltrate the cache built from the initial samples, causing it to forget previously learned information. In other words, the cache updated with Sequence A represents the state **before forgetting**, while the cache updated with Sequence B represents the state **after forgetting**. To quantify the impact of forgetting, we measured the performance difference between Sequence A and Sequence B on the final test samples. The performance **drop** is calculated as (**Accuracy of A - Accuracy of B**). The results are shown in the table below:
>
> | Drop (↓) | Aircraft | Caltech | Cars | DTD  | EuroSAT | Flower | Food101 | Pets | SUN397 | UCF101 | Sum  |
> | -------- | :------: | :-----: | :--: | :--: | :-----: | :----: | :-----: | :--: | :----: | :----: | :--: |
> | DPE      |   0.32   |  0.08   | 0.24 | 0.35 |  1.03   |  0.33  |  0.12   | 0.73 |  0.16  |  0.34  | 3.7  |
> | SCA      |    0     |  -0.34  | 0.12 | 0.26 |    0    | -0.27  |  0.08   | 0.47 |   0    | -0.42  | -0.1 |
>
> We have several observations: (1) The SOTA baseline DPE is highly vulnerable to forgetting. (2) SCA is inherently robust to forgetting, as its accumulation design prevents error samples from fully overwriting the stored knowledge. (3) Remarkably, SCA even leverages these "forgetting triggers" to improve performance (a negative drop), turning a vulnerability into a strength.
>
> **2.Blocking issues**
>
> We present additional results about the blocking issues through simulating "bad cases":  we randomly selected 10 classes and identified the single lowest-entropy misclassified sample for each class, placing them at the beginning of the test sample sequence. As shown in the table below, the impact of blocking issues on SCA's performance is much smaller that DPE. This is because SCA's statistics-based accumulation is not limited by fixed cache slots, preventing initial high-confidence errors from occupying resources and hindering learning from subsequent samples.
>
> |                 | Aircraft | Caltech | Cars  |  DTD  | EuroSAT | Flower | Food101 | Pets  | SUN397 | UCF101 | Average |
> | --------------- | :------: | :-----: | :---: | :---: | :-----: | :----: | :-----: | :---: | :----: | :----: | :-----: |
> | DPE (bad case)  |  27.69   |  92.24  | 66.36 | 52.62 |  54.54  | 73.65  |  85.98  | 90.28 | 68.82  | 70.08  |  68.23  |
> | SCA (bad case)  |  28.49   |  93.23  | 67.88 | 55.91 |  55.72  | 75.64  |  86.18  | 91.41 | 70.34  | 73.20  |  69.81  |
> | DPE (good case) |  28.95   |  94.81  | 67.31 | 54.20 |  55.79  | 75.07  |  86.17  | 91.14 | 70.07  | 70.44  |  69.40  |
> | SCA (good case) |  28.50   |  94.85  | 68.49 | 57.09 |  57.16  | 76.09  |  86.09  | 91.44 | 70.27  | 73.43  |  70.34  |
>
> Finally, we want to clarify the practical relevance of these controlled experiments. These forgetting and blocking issues are not artifacts of our experimental design. On the contrary, they are constantly occurring at a micro-level in any test sequence. However, the negative impact of bad cache updates (forgetting and blocking issues) may be **obscured** by the effects of subsequent good updates. This makes it hard to directly observe or quantify the harm of these vulnerabilities in a standard evaluation. The purpose of our controlled experiments is to isolate these specific variables. By altering the sequence of just a small fraction of the data, we create a scenario where the harm from forgetting and blocking can be directly and intuitively measured.
>
> We will update the corresponding content (e.g., Figure 2) in the original paper based on the existing content and include additional visualizations in the final version.

---

### Official Review · Reviewer_GK6p · 2025-07-02

**Clarity:** 3
**Significance:** 3
**Originality:** 3
**Rating:** 5
**Confidence:** 5

**Summary:**

This paper introduces Statistics Caching Test-Time Adaptation (SCA), a training-free test-time adaptation method for CLIP that replaces fixed-size caches with running class-wise statistics. Instead of storing features, it updates class means using soft pseudo-labels weighted by confidence. Predictions are made by combining this stats-based classifier with CLIP’s original output. The method avoids cache issues like forgetting or blocking and shows strong accuracy gains across benchmarks with the adaptive instance-level fusion strategy.

**Questions:**

1. Lines 176–177 state that allowing |Ω| to vary with τ and the sample’s confidence results in softer pseudo-labels for uncertain samples and sharper ones when confident. Could you provide more evidence for this claim, either empirically or through visualization?

2. The results on ImageNet with the ResNet backbone appear weaker than DPE. Could you clarify why your method underperforms in this setting and whether it's due to architecture sensitivity or some other factor?

3. In your efficiency comparison (Table 3), do the reported numbers include the cost of data augmentation? If not, could you provide a breakdown to help interpret the practical runtime tradeoffs?

4. How did you determine the default values for your hyperparameters? Were they tuned on a subset of data, or selected based on prior work?

5. Why are there no ablation studies on ImageNet? Given its scale and diversity, this would provide a more rigorous validation than DTD, which is relatively small.

6. How would you explain the performance drop between Table 1 and Table 2? Is it due to the number of classes, the nature of the domain shift, or semantic similarity in CLIP’s textual space? Some analysis here would help interpret the results.

7. Lines 242–244 present an important insight. Could you consider highlighting this point earlier or making it more prominent in the Table 3 caption, for instance? It feels like a core comparison factor for those who are not familiar with DPE and might be overlooked.

If my concerns are addressed, I will raise my score.

**Ethical Concerns:**

["NO or VERY MINOR ethics concerns only"]

**Final Justification:**

Thanks to the authors for the detailed response. As I said, I’m updating my score to "Accept" since all concerns were resolved. If the paper gets accepted, please make sure to include the clarifications and changes I suggested. I also encourage open-sourcing the project to make it accessible to others.

**Limitations:**

The method introduces some storage overhead due to maintaining feature statistics. It partially mitigates error accumulation, but in settings with low domain shift, the performance gains seem to become smaller.

**Paper Formatting Concerns:**

The format is matched

**Quality:**

3

**Strengths And Weaknesses:**

**Strengths:**
1. The motivation is clear, and has a well-structured presentation with Figure 1.
2. Replaces fixed-size caches with statistical summaries, directly addressing forgetting and blocking issues in cache-based adaptation.
3. SCA outperforms prior methods like TDA and DPE across multiple benchmarks, showing robustness with the proposed hyperparameter strategy picking.
4. Computationally efficient: the method avoids optimization and maintains low overhead per sample, with inference time comparable to other lightweight approaches.

**Weaknesses:**
1. The results on ResNet backbone with OOD benchmark is weaker than previous SOTA DPE.
2. No ablations on bigger scope dataset like ImageNet, small-scale datasets like DTD is used for ablation studies.
3. Efficiency comparisons lack clarity, unclear whether data augmentation cost is included.

---

> ### Author Rebuttal · Authors · 2025-07-31
>
> We would like to thank the reviewer for taking the time to review our work. We appreciate that you find our paper well-motivated. According to your valuable comments, we provide detailed feedback.
> ### **W1 & Q2 & Q6: Performance on OOD benchmark**
>
> Thank you for your insightful question regarding the performance comparison with the SOTA baseline DPE [1] on the OOD benchmark using a ResNet-50 backbone. Here we provide the explanation:
>
> Our proposed SCA is an entirely **training-free** approach. It operates by accumulating feature statistics (first and second-order moments) from all past samples and constructs a classifier via a **closed-form solution** (as detailed in Eq. 6, 7, and 8). The performance of this design is therefore directly and heavily dependent on the intrinsic quality and linear separability of the features provided by the backbone network.
>
> The OOD benchmark is a particularly demanding scenario; its large number of classes and fine-grained **stylistic shifts** place extreme requirements on the discriminative power of the features. That's why performance drop occurs between Cross-Domain benchmark and OOD benchmark **(Q6: Performance drop between Table 1 and Table 2)**.
>
> While ResNet is a powerful CNN architecture, its features may be less semantically rich or linearly separable compared to those from ViT backbones. In this context, DPE's additional training phase can effectively compensate for minor imperfections in the ResNet-50 feature space, allowing it to find a slightly better decision boundary and thus achieve a marginally higher accuracy. Our purely statistical SCA is more constrained by the inherent quality of these features.
>
> [1] Dual Prototype Evolving for Test-Time Generalization of Vision-Language Models, NeurIPS'24
>
> ### **W2 & Q5: Ablation on ImageNet**
>
> Thank you for this suggestion. In our paper, our ablation studies were primarily conducted on the cross-domain benchmark for faster validation. Here, we provide the results of ablation study on ImageNet:
>
> | Method Configuration               | ImageNet Accuracy |
> | :--------------------------------- | :---------------: |
> | w/o statistics accumulation (M=4)  |       70.79       |
> | w/o statistics accumulation (M=8)  |       70.81       |
> | w/o statistics accumulation (M=16) |       71.02       |
> | w/o statistics accumulation (M=32) |       71.08       |
> | w/o dynamic soft label assignment  |       70.65       |
> | **SCA (with both)**                |     **71.75**     |
>
> As shown in table, we have several observations: (1) our full method (`71.75%`) consistently outperforms all variants of the feature caching baseline (`w/o statistics accumulation`). This demonstrates that our approach of accumulating knowledge from all samples into compact statistics is fundamentally more effective than maintaining a limited-size cache of discrete features, even on large-scale datasets. (2) Dynamic soft label assignment plays a great role in performance gain. This underscores the critical importance of this component, especially on a challenging, large-scale dataset with 1000 classes where the risk of generating incorrect hard pseudo-labels is extremely high. This confirms that our uncertainty-aware labeling is essential for robust adaptation.
>
> ### **W3 & Q3: Efficiency comparison (whether data augmentation cost is included)**
>
> Thank you for this important question regarding the efficiency comparison.
>
> The numbers reported in Table 3 exclude data augmentation, as none of the methods use it on the cross-domain benchmark. This is a standard evaluation protocol followed by prior work in this area, including the baselines we compare against like DPE and TDA. Therefore, the reported runtimes reflect the full processing time for each method under this no-augmentation setting.
>
> ### **Q1: Varying $|\Omega|$ and its impact on pseudo-label sharpness**
>
> Thanks for your insightful question. The core logic behind this claim is tied to how the candidate set $\Omega$ is formed based on the initial CLIP prediction probabilities ($p_\mathrm{text}$) and the threshold $\tau(1-p_\mathrm{text}^{(1)})$. Here $p_\mathrm{text}^{(1)}$ is the value of the top-ranked class in $p_\mathrm{text}$.
>
> * **For a confident sample:** The initial prediction $p_\mathrm{text}$ may exhibit a very high probability for one class ($p_\mathrm{text}^{(1)}$). According to Eq. (9), the candidate set $\Omega$ will likely contain only a small number of classes (small $|\Omega|$) that exceed the threshold $\tau(1-p_\mathrm{text}^{(1)})$. Upon renormalization in Eq. (10), the resulting pseudo-label $\hat{Y}_t$ becomes a near one-hot vector, making it **sharp**.
> * **For an uncertain sample:** The initial prediction $p_\mathrm{text}$ shows probabilities spread across multiple classes, none of which is individually large. To meet the cumulative probability threshold $\tau(1 - p_\mathrm{text}^{(1)})$, Eq. (9) includes multiple classes in the candidate set $\Omega$ (resulting in a large $|\Omega|$). As a result, the final pseudo-label $\hat{Y}_t$ distributes its probability mass across these candidates, making it **soft.**
>
> Compared to using hard labels, which completely ignore uncertainty information, our dynamic threshold provides a way to adapt the label's sharpness based on the model's own confidence.
>
> ### **Q4: Hyperparameters selection**
>
> We select hyperparameters based on the validation set of DTD, a relatively challenging dataset (with CLIP ViT zero-shot accuracy of 44.27). Additionally, DTD has a small number of test samples, which allows for efficient validation. That's why DTD appears frequently across multiple tables.
>
> ### **Q7:  Suggestion on highlighting "whether need training" in Table 3**
>
>
> Thanks for your insightful suggestion, which helps make our paper clearer. In the final version, we will add a "Requires Training" column to Table 3 to highlight that our method, unlike DPE, is training-free.
>
> We deeply appreciate your efforts and would be happy to further clarify or discuss any remaining concerns. Please feel free to let us know if additional modifications or explanations are required.

---

> > ### Comment · Reviewer_GK6p · 2025-08-05
> > **Reviewer Reply to Rebuttal**
> >
> > Thank you for the rebuttal and the additional experiments. These clarifications addressed the main issues I raised, and I have adjusted my score to reflect my updated evaluation.

---

> > > ### Author Response · Authors · 2025-08-06
> > >
> > > We sincerely appreciate your thoughtful feedback and support, and will make the corresponding revisions in the final version.

---

### Decision · Program_Chairs · 2025-09-17

**Decision:**

Accept (poster)

**Comment:**

This paper proposes Statistics Caching Test-Time Adaptation (SCA), a training-free method for CLIP. SCA replaces fixed caches with running class-wise statistics, thereby avoiding issues such as forgetting and blocking in prior methods, while also achieving good performance.

After multiple rounds of discussion, the authors provided further clarifications and experimental results to support their claims. The reviewers agreed that the concerns were resolved and reached a consensus to accept the paper. The AC also endorsed this decision and recommended acceptance.

Kindly ensure that the new rebuttal results and discussions are incorporated into the final version.